# On the Implicit Bias Towards Depth Minimization in Deep Neural Networks

## Abstract

Recent results in the literature suggest that the penultimate (second-to-last) layer representations of neural networks that are trained for classification exhibit a clustering property called neural collapse (NC). We study the implicit bias of stochastic gradient descent (SGD) to favor of low-depth solutions when training deep neural networks. We characterize a notion of effective depth that measures the first layer for which sample embeddings are separable using the nearest-class center classifier. Furthermore, we hypothesize and empirically show that SGD implicitly selects neural networks of small effective depths.

Secondly, while neural collapse emerges even when generalization should be impossible - we argue that the *degree of separability* in the intermediate layers is related to generalization. We derive a generalization bound based on comparing the effective depth of the network with the minimal depth required to fit the same dataset with partially corrupted labels. Remarkably, this bound provides non-trivial estimations of the test performance and is independent of the depth.

## 1 Introduction

Deep learning systems have steadily advanced the state of the art in a wide range of benchmarks, demonstrating impressive performance in tasks ranging from image classification (Taigman et al., 2014; Zhai et al., 2021), language processing (Devlin et al., 2019; Brown et al., 2020), open-ended environments (Silver et al., 2016; Arulkumaran et al., 2019), to coding (Chen et al., 2021).

Recent research indicates that deep neural networks generalize well, in part because the number of parameters exceeds the number of training samples (Belkin et al., 2018; Belkin, 2021; Advani & Saxe, 2017; Belkin et al., 2019). However, it has been shown that in this case deep learning models can precisely interpolate arbitrary training labels (Zhang et al., 2017) (also known as the "interpolation regime"). Therefore, understanding interpolation learning appears to be a critical step toward a better theoretical understanding of deep learning's successes.

Traditional generalization bounds (Vapnik, 1998; Shalev-Shwartz & Ben-David, 2014; Mohri et al., 2012; Bartlett & Mendelson, 2003) are based on uniform convergence. In this approach, instead of directly analyzing the population error of a learning algorithm, a uniform convergence-type argument would control the worst-case generalization gap (distance between train and test errors) over a class of predictors containing the outputs of the learning algorithm. Typically, this is done because for many algorithms it is difficult to exactly characterize the learned predictor.

Nagarajan & Kolter (2019), however, raised significant questions about the applicability of typical uniform convergence arguments to certain interpolation learning regimes. They described theoretical settings in which an interpolation learning algorithm generalizes well but any uniform convergence bound cannot identify that. Following their work, Bartlett & Long (2021); Zhou et al. (2020); Negrea et al. (2020); Yang et al. (2021) all demonstrated the failure of forms of uniform convergence in various interpolation learning setups.

**Contributions.** Because of the inherent limitations of uniform convergence bounds, in this paper we pursue a novel approach for measuring generalization in deep learning that is not based on uniform convergence. Instead, our bound suggests that the model performs well at test time if its complexity is small *compared* to the complexity of a network required to fit the same dataset with partially random labels. In other words, even if a trained network has a complexity greater than the

| Dataset | MNIST | | | Fashion MNIST | | | CIFAR10 | | |
|---|---|---|---|---|---|---|---|---|---|
| Architecture | CONV-$L$-50 | | | CONV-$L$-100 | | | CONV-$L$-100 | | |
| Depth ($L$) | 10 | 12 | 15 | 10 | 12 | 15 | 16 | 18 | 20 |
| Test error | 0.0075 | 0.0074 | 0.0074 | 0.0996 | 0.0996 | 0.0996 | 0.2659 | 0.2653 | 0.2648 |
| $p$ | 0.1 | 0.1 | 0.1 | 0.2 | 0.2 | 0.2 | 0.4 | 0.4 | 0.4 |
| Our bound | **0.1** | **0.1** | **0.1** | **0.2** | **0.2** | **0.2** | **0.66** | **0.66** | **0.53** |
| $L_{1,\infty}$ (Bartlett & Mendelson, 2003) | 8.911e+14 | 1.74e+17 | 2.13e+22 | 3.613e+17 | 9.145e+18 | 4.088e+22 | 1.076e+23 | 6.682e+28 | 2.758e+35 |
| $L_{3,1.5}$ (Neyshabur et al., 2015) | 5.462e+05 | 1.6e+06 | 1.308e+06 | 7.523e+07 | 6.997e+07 | 2.636e+08 | 4.633e+08 | 2.275e+09 | 5.061e+09 |
| Frobenius (Neyshabur et al., 2015) | 1.848e+06 | 8.194e+06 | 2.216e+07 | 2.486e+08 | 2.335e+08 | 1.585e+09 | 1.967e+09 | 1.442e+10 | 3.038e+11 |
| Spec $L_1$ (Bartlett et al., 2017) | 2.861e+05 | 6.412e+05 | 9.566e+05 | 4.706e+06 | 3.516e+06 | 3.176e+06 | 1.19e+07 | 1.449e+08 | 1.272e+10 |
| Spec Frob (Neyshabur et al., 2019) | 3.948e+03 | 1.1199e+04 | 1.538e+04 | 4.0229e+04 | 2.884e+04 | 2.543e+04 | 9.4833e+04 | 1.011e+06 | 1.033e+08 |

Table 1: **Comparing our bound with baseline bounds in the literature for networks of varying depths.** Our error bound is reported in the fourth row, and the baseline bounds are reported in the bottom rectangle. While the test error is universally bounded by 1, the baseline bounds are much larger than 1, and therefore, are meaningless. **In contrast, our bound achieves relatively tight estimations of the test error and unlike the baseline bounds, our bound is fairly unaffected by the network's depth.**

number of training samples, it may be less complex than a model that fits partially random labels. As a result, in such cases, our bound may provide a non-trivial estimate of the test error.

To formally describe our notion of complexity, we employ the notion of nearest class-center (NCC) separability. This property asserts that the feature embeddings associated with training samples belonging to the same class are separable according to the nearest class-center decision rule. While original results (Papyan et al., 2020) observed NCC separability at the penultimate layer of trained networks, recent results (Ben-Shaul & Dekel, 2022) observed NCC separability also in intermediate layers. In this work, we introduce the notion of *'effective depth'* of neural networks that regards to the lowest layer for which its features are NCC separable (see Sec. 3.2).

We make multiple important observations regarding effective depths. **(i)** We empirically show that the effective depth of trained networks monotonically increases when increasing the amount of random labels in data. **(ii)** We observe that when training sufficiently deep networks, they converge to (approximately) the same effective depth $L_0$. Furthermore, as we show in Tab. 1, unlike traditional generalization bounds, **our bound is empirically non-vacuous and independent of depth**.

## 1.1 RELATED WORK

**Neural collapse and generalization.** Our work is closely related to the recent line of work on Neural collapse (Papyan et al., 2020; Han et al., 2022). Neural collapse identifies training dynamics of deep networks for standard classification tasks, where the feature embeddings associated with training samples belonging to the same class tend to concentrate around their means.

While several papers analyzed the emergence of neural collapse from a theoretical standpoint (e.g., (Zhu et al., 2021; Rangamani et al., 2022; Lu & Steinerberger, 2020; Fang et al., 2021; Ergen & Pilanci, 2021)), its specific role in deep learning and its potential relationship with generalization is still unclear. Recent work (Galanti et al., 2022a; Xu et al., 2022; Galanti et al., 2022b) studied the conditions for when class features variation collapse generalizes from the train samples, to both test samples and new classes and in the transfer learning setting.

We focus on the following (independent) question in this work: *Is neural collapse a good indicator of how well a network generalizes?* As a counter-argument, Zhu et al. (2021) provided empirical evidence that neural collapse occurs even when training the network with random labels. As a result, the presence of neural collapse cannot indicate whether or not the network generalizes. This experiment, however, does not rule out the possibility of an indirect relationship between neural collapse and generalization. We contend that the degree of separability in the intermediate layers is related to generalization.

**Emergence of structure in deep networks.** While various papers Papyan (2020); Tirer & Bruna (2022); Galanti et al. (2022a); Ben-Shaul & Dekel (2022); Cohen et al. (2018); Alain & Bengio (2017); Montavon et al. (2011); Papyan et al. (2017); Ben-Shaul & Dekel (2021); Shwartz-Ziv & Tishby (2017) investigated certain geometrical properties within intermediate layers (e.g., clustering and separability), this paper is the first to demonstrate that deep neural networks tend to converge to a minimal effective depth that is independent of the network's depth. Even though one can de-

rive "effective depths" from the experiments of Cohen et al. (2018), we show that when training sufficiently deep networks they converge to (approximately) the same effective depth.

## 2 PROBLEM SETUP

In this section we describe the learning setting we use in the theory and experiments. We consider the problem of training a model for standard multi-class classification. Formally, the target task is defined by a distribution $P$ over samples $(x, y) \in \mathcal{X} \times \mathcal{Y}_C$, where $\mathcal{X} \subset \mathbb{R}^d$ is the instance space, and $\mathcal{Y}_C$ is a label space with cardinality $C$. To simplify the presentation, we use one-hot encoding for the label space, that is, the labels are represented by the unit vectors in $\mathbb{R}^C$, and $\mathcal{Y}_C := \{e_c : c = 1, \ldots, C\}$ where $e_c \in \mathbb{R}^C$ is the $c$th standard unit vector in $\mathbb{R}^C$; with a slight abuse of notation, we allow ourselves to write $y = c$ instead of $y = e_c$. For a pair $(x, y)$ distributed by $P$, we denote by $P_c$ the class conditional distribution of $x$ given $y = c$ (i.e., $P_c(\cdot) := \mathbb{P}[x \in \cdot \mid y = c]$).

A classifier $h_W : \mathcal{X} \to \mathbb{R}^C$ assigns a *soft* label to an input point $x \in \mathcal{X}$, and its performance on the distribution $P$ is measured by the expected risk

$$L_P(h_W) := \mathbb{E}_{(x, y(x)) \sim P}[\ell(h_W(x), y(x))],$$

where $\ell : \mathbb{R}^C \times \mathcal{Y}_C \to [0, \infty)$ is a non-negative loss function (e.g., $L_2$ or cross-entropy losses).

We typically do not have direct access to the full population distribution $P$. Therefore, we generally aim to learn a classifier, $h$, using some balanced training data $S := \{(x_i, y_i)\}_{i=1}^m = \cup_{c=1}^C S_c = \cup_{c=1}^C \{x_{ci}, y_{ci}\}_{i=1}^{m_0} \sim P_B(m)$ of $m = C \cdot m_0$ samples consisting $m_0$ independent and identically distributed (i.i.d.) samples drawn from $P_c$ for each $c \in [C]$. Specifically, we intend to find $W$ that minimizes the regularized empirical risk

$$L_S^\lambda(h_W) := \frac{1}{m} \sum_{i=1}^m \ell(h_W(x_i), y_i) + \lambda \|W\|_2^2, \tag{1}$$

where the regularization controls the complexity of the function $h_W$ and typically helps reducing overfitting. Finally, the performance of the trained model is evaluated using the train and test error rates; $\mathrm{err}_S(h_W) := \sum_{i=1}^m \mathbb{I}[\arg\max_c h_W(x_i)_c \neq y_i]$ and $\mathrm{err}_P(h_W) := \mathbb{E}_{(x,y) \sim P}[\mathbb{I}[\arg\max_c h_W(x)_c \neq y]]$, where $\mathbb{I} : \{\mathrm{True}, \mathrm{False}\} \to \{0, 1\}$ the indicator function.

**Neural networks.** In this work, the classifier $h_W$ is a neural network, decomposed into a set of parametric layers. Formally, we write $h_W := e_{W_e} \circ f_{W_f}^L := e_{W_e} \circ g_{W_L}^L \circ \cdots \circ g_{W_1}^1$, where $g_{W_i}^i \in \{g' : \mathbb{R}^{p_i} \to \mathbb{R}^{p_{i+1}}\}$ are parametric functions and $e_{W_e} \in \{e' : \mathbb{R}^{p_{L+1}} \to \mathbb{R}^C\}$ is a linear function. For example, $g_{W_i}^i$ could be a standard linear or convolutional layer, a residual block or a pooling layer. Here, $\sigma$ is an element-wise ReLU activation function. With a slight abuse of notation, we omit specifying the sub-scripted weights, $f_i := g^i \circ \cdots \circ g^1$ and $h := h_W$.

**Optimization.** We optimize our models to minimize the regularized empirical risk $L_S^\lambda(h)$ by applying SGD for a certain number of iterations $T$ with coefficient $\lambda > 0$. Specifically, we initialize the weights $W_0 = \gamma$ of $h$ using a standard initialization procedure and at each iteration, we update $W_{t+1} \leftarrow W_t - \mu_t \nabla_W L_{\tilde{S}}(h_t)$, where $\mu_t > 0$ is the learning rate at the $t$'th iteration and the subset $\tilde{S} \subset S$ of size $B$ is selected uniformly at random. Throughout the paper, we denote by $h_S^\gamma$ the output of the learning algorithm starting from the initialization $W_0 = \gamma$. When $\gamma$ is irrelevant or obvious from context, we will simply write $h_S^\gamma = h_S = e_S \circ f_S$.

## 3 NEURAL COLLAPSE AND GENERALIZATION

In this section we theoretically explore the relationship between neural collapse and generalization. We start by introducing neural collapse, NCC separability, and effective depth of neural networks. Then, we connect these notions with the test-time performance of neural networks.

### 3.1 NEAREST CLASS-CENTER SEPARABILITY

Neural collapse identifies training dynamics of deep networks for standard classification tasks, in which the features of the penultimate layer associated with training samples belonging to the same

class tend to concentrate around their class-means. This includes (NC1) class-features variability collapse, (NC2) the class means of the embeddings collapse to the vertices of a simplex equiangular tight frame, (NC3) the last-layer classifiers collapse to the class means up to scaling and (NC4) the classifier's decision collapses to simply choosing whichever class has the closest train class mean, while maintaining a zero classification error.

In this paper we focus on a weak form of NC4 we call *"nearest class-center separability"* (NCC separability). Formally, suppose we have a dataset $S = \cup_{c=1}^{C} S_c$ of samples and a mapping $f : \mathbb{R}^d \to \mathbb{R}^p$, the features of $f$ are NCC separable (w.r.t. $S$) if for all $i \in [m]$, we have $\hat{h}(x_i) = y_i$, where $\hat{h}(x) := \arg\min_{c \in [C]} \|f(x) - \mu_f(S_c)\|$. To measure the degree of NCC separability of a feature map $f$, we use the train and test classification error rates of the NCC classifier on top of the given layer, $\mathrm{err}_S(\hat{h})$ and $\mathrm{err}_P(\hat{h})$.

Essentially, NC4 asserts that during training, the feature embeddings in the penultimate layer become separable and the classifier $h$ itself converges to the 'nearest class-center classifier' $\hat{h}$.

## 3.2 Effective Depths and Generalization

In this section we study the effective depths of neural networks and their connection with generalization. To formally define this notion, we focus on neural networks whose $L$ top-most layers are of the same size. We observe that neural networks trained for standard classification exhibit an implicit bias towards depth minimization.

**Observation 1** (Minimal depth hypothesis). *Suppose we have a dataset $S$. There exists an integer $L_0 \geq 1$, such that, if we train a neural network of any depth $L \geq L_0$ for cross-entropy minimization on $S$ using SGD with weight decay, the learned features $f^l$ become (approximately) NCC separable for all $l \in \{L_0, \ldots, L\}$.*

We note that if the $L_0$'th layer of $f_L$ exhibits NCC separability, we could correctly classify the samples already in the $L_0$'th layer of $f_L$ using a linear classifier (i.e., the nearest class-center classifier). Therefore, intuitively its depth is effectively upper bounded by $L_0$. The notion of effective depth of a neural network is formally defined as follows.

**Definition 1** ($\epsilon$-effective depth). *Suppose we have a dataset $S$ and a neural network $h = e \circ g^L \circ \cdots \circ g^1$ with $g^1 : \mathbb{R}^n \to \mathbb{R}^{p_2}$, $g^i : \mathbb{R}^{p_i} \to \mathbb{R}^{p_{i+1}}$ and linear classifier $e : \mathbb{R}^{p_{L+1}} \to \mathbb{R}^C$. Let $\hat{h}_i(x) := \arg\min_{c \in [C]} \|f_i(x) - \mu_{f_i}(S_c)\|$. The $\epsilon$-effective depth $\ell_S^\epsilon(h)$ of the network $h$ is the minimal value $i \in [L]$, such that, $\mathrm{err}_S(\hat{h}_i) \leq \epsilon$ (and $\ell_S^\epsilon(h) = L$ if such $i \in [L]$ is non-existent).*

To avoid confusion, we note that the $\epsilon$-effective depth is a property of a neural network and not of the function it implements. That is, a function can be implemented by two different architectures of different effective depths. While our empirical observations in Sec. 4 suggest that the optimizer learns neural networks of low-depths, it is not necessarily the lowest depth that allows NCC separability. As a next step, we define the $\epsilon$-*minimal NCC depth*. Intuitively, the NCC depth of a given architecture is the minimal value $L \in \mathbb{N}$, for which there exists a neural network of depth $L$ whose features are NCC separable. As we will show, the relationship between the $\epsilon$-effective depth of a neural network and the $\epsilon$-minimal NCC depth is connected with generalization.

**Definition 2** ($\epsilon$-Minimal NCC depth). *Suppose we have a dataset $S = \cup_{c=1}^{C} S_c$ and a neural network architecture $f^L = g^L \circ \cdots \circ g^1$ with $g^1 : \mathbb{R}^n \to \mathbb{R}^{n_0}$ and $g^i \in \mathcal{G} \subset \{g' \mid g' : \mathbb{R}^{n_0} \to \mathbb{R}^{n_0}\}$ for all $i = 2, \ldots, L$. The $\epsilon$-minimal NCC depth of $\mathcal{G}$ is the minimal depth $L$ for which there exist parameters $W = \{W_i\}_{i=1}^{L}$, such that, $f' := f_W^L = g_{W_L}^L \circ \cdots \circ g_{W_1}^1$ satisfies $\mathrm{err}_S(\hat{h}) \leq \epsilon$, where $\hat{h}(x) := \arg\min_{c \in [C]} \|f'(x) - \mu_{f'}(S_c)\|$. We denote the $\epsilon$-minimal NCC depth by $\ell_{\min}^\epsilon(\mathcal{G}, S)$.*

To study the performance of a given model, we consider the following setup. Let $S_1 = \{(x_i^1, y_i^1)\}_{i=1}^{m}$ and $S_2 = \{(x_i^2, y_i^2)\}_{i=1}^{m}$ be two balanced datasets. We think of them as two splits of the training dataset $S$. We assume that the classifier $h_{S_1}^\gamma$ is trained on $S_1$ and we use $S_2$ to evaluate its performance. We denote by $X_j = \{x_i^j\}_{i=1}^{m}$ and $Y_j = \{y_i^j\}_{i=1}^{m}$ the instances and labels in $S_j$.

To formally state our bound, we make two technical assumptions. The first is that the misclassified labels that $h_{S_1}^\gamma$ produces over the samples $X_2 = \cup_{c=1}^{C} \{x_{ci}^2\}_{i=1}^{m_0}$ are distributed uniformly.

**Definition 3** ($\delta_m$-uniform mistakes). *We say that the mistakes of a learning algorithm $A : (S_1, \gamma) \mapsto h_{S_1}^\gamma$ are $\delta_m$-uniform, if with probability $\geq 1 - \delta_m$ over the selection of $S_1, S_2 \sim P_B(m)$, the values and indices of the mistaken labels of $h_{S_1}^\gamma$ over $X_2$ are uniformly distributed (as a function of $\gamma$).*

The above definition provides two conditions regarding the learning algorithm. It assumes that with a high probability (over the selection of $S_1, S_2$), $h_{S_1}^\gamma$ makes the same number of mistakes on $S_2$ across all initializations $\gamma$. In addition, it assumes that the mistakes are distributed uniformly across the samples in $S_2$ and their (incorrect) values are also distributed uniformly. While these assumptions may be violated in practice, the train error typically has a small variance and the mistakes are almost distributed uniformly when the classes are non-hierarchical (e.g., CIFAR10, MNIST).

For the second assumption, we consider the following term. Let $p \in (0, 1/2), \alpha \in (0, 1)$, we denote

$$\delta_{m,p,\alpha}^2 := \mathbb{P}_{S_1, S_2, \tilde{Y}_2, \hat{Y}_2} \left[ \exists\, q \geq (1 + \alpha)\, p : \mathscr{d}_{\min}^\epsilon(\mathcal{G}, S_1 \cup \tilde{S}_2) > \mathbb{E}_{\hat{Y}_2}[\mathscr{d}_{\min}^\epsilon(\mathcal{G}, S_1 \cup \hat{S}_2)] \right], \quad (2)$$

where $\tilde{Y}_2 = \{\tilde{y}_i\}_{i=1}^m$ and $\hat{Y}_2 = \{\hat{y}_i\}_{i=1}^m$ are uniformly selected to be sets of labels that disagree with $Y_2$ on $pm$ and $qm$ values (resp.) and $\tilde{S}_2$ and $\hat{S}_2$ are datasets obtained by replacing the labels of $S_2$ with $\tilde{Y}_2$ and $\hat{Y}_2$ (resp.). We assume that $\delta_{m,p,\alpha}^2$ is small. Meaning, with a high probability, the minimal depth to fit $(2 - p)m$ correct labels and $pm$ random labels is upper bounded by the expected minimal depth to fit $(2 - q)m$ correct labels and $qm$ random labels for any $q \geq (1 + \alpha)p$. To understand this assumption, we note that in both cases, the model has to fit at least $m$ correct labels and $pm$ (or $qm$) random labels. However, we typically need to increase the capacity of the model in order to fit extended amounts of random labels (see Figs. 3).

Following the setting above, we are prepared to formulate our generalization bound.

**Proposition 1.** *Let $m \in \mathbb{N}$, $p \in (0, 1/2)$, $\alpha \in (0, 1)$ and $\epsilon \in (0, 1)$. Assume that the error of the learning algorithm is $\delta_m^1$-uniform. Assume that $S_1, S_2 \sim P_B(m)$. Let $h_{S_1}^\gamma$ be the output of the learning algorithm given access to a dataset $S_1$ and initialization $\gamma$. Then,*

$$\begin{aligned} \mathbb{E}_{S_1} \mathbb{E}_\gamma[\mathrm{err}_P(h_{S_1}^\gamma)] &\leq \mathbb{P}_{S_1, S_2, \tilde{Y}_2} \left[ \mathbb{E}_\gamma[\mathscr{d}_{S_1}^\epsilon(h_{S_1}^\gamma)] \geq \mathscr{d}_{\min}^\epsilon(\mathcal{G}, S_1 \cup \tilde{S}_2) \right] \\ &\quad + (1 + \alpha)p + \delta_m^1 + \delta_{m,p,\alpha}^2, \end{aligned} \quad (3)$$

*where $\tilde{Y}_2 = \{\tilde{Y}_i\}_{i=1}^m$ is uniformly selected to be a set of labels that disagrees with $Y_2$ on $pm$ values.*

The above proposition provides an upper bound on the expected test error of the classifier $h_{S_1}^\gamma$ which is the term that we would like to bound. The proposition assumes that the mistakes $h_{S_1}^\gamma$ generates on $X_2$ are distributed uniformly (with probability $\geq 1 - \delta_m^1$). To account the likelihood that this assumption fails, our bound includes the term $\delta_m^1$, which is assumed to be small.

Informally, the bound suggests the following idea to evaluate the performance of $h_{S_1}^\gamma$. We start with an initial guess $p_m = p \in (0, 1/2)$ of the test error of $h_{S_1}^\gamma$. Using this guess, we compare its $\epsilon$-effective depth with the $\epsilon$-minimal NCC depth $\mathscr{d}_{\min}^\epsilon(\mathcal{G}, S_1 \cup \tilde{S}_2)$ required to NCC separate the samples in $S_1 \cup \tilde{S}_2$, where $\tilde{S}_2$ is the result of randomly relabeling $p_m m$ of $S_2$'s labels. Intuitively, if the mistakes of $h_{S_1}^\gamma$ are uniformly distributed and its $\epsilon$-effective depth is smaller than $\mathscr{d}_{\min}^\epsilon(\mathcal{G}, S_1 \cup \tilde{S}_2)$, then, we expect $h_{S_1}^\gamma$ to make at most $p_m$ mistakes on $S_2$. Therefore, in a sense, the choice of $p_m$ serves as a 'guess' whether the effective depth of a model trained with $S_1$ is likely to be smaller than the $\epsilon$-minimal NCC depth required to NCC separate the samples in $S_1 \cup \tilde{S}_2$.

Next, we interpret each term separately. The term $\mathbb{E}_\gamma[\mathscr{d}_{S_1}^\epsilon(h_{S_1}^\gamma)]$ depends on the complexity of the classification problem and the implicit bias of SGD to favor networks of small $\epsilon$-effective depths. In the worst case, if SGD does not minimize the $\epsilon$-effective depth or the labels in $S_1$ are random (and $m$ is sufficiently large), we expect $\mathbb{E}_\gamma[\mathscr{d}_{S_1}^\epsilon(h_{S_1}^\gamma)] = L$. On the other hand, $\mathscr{d}_{\min}^\epsilon(\mathcal{G}, S_1 \cup \tilde{S}_2)$ measures the complexity of a task that involves fitting a dataset of size $2m$ samples, where $(2 - p_m)m \geq m$ of the labels are correct and $p_m m$ are random labels. By decreasing $p_m$, we expect $\mathscr{d}_{\min}^\epsilon(\mathcal{G}, S_1 \cup \tilde{S}_2)$ to decrease, making the first term in the bound larger. In addition, if $h = e \circ f^L$ is a neural network of a fixed width, it is impossible to fit an increasing amount of random labels without increasing the depth. Therefore, when $p_m m \xrightarrow{m \to \infty} \infty$, the dataset $S_1 \cup \tilde{S}_2$ becomes increasingly harder to fit, and

we expect $\ell_{\min}^{\epsilon}(\mathcal{G}, S_1 \cup \tilde{S}_2)$ to tend to infinity. If $\mathbb{E}_{\gamma}[\ell_{S_1}^{\epsilon}(h_{S_1}^{\gamma})]$ is bounded as a function of $L$ and $m$ and if $p_m m \xrightarrow[m \to \infty]{} \infty$, we obtain that $\mathbb{P}[\mathbb{E}_{\gamma}[\ell_{S_1}^{\epsilon}(h_{S_1}^{\gamma})] \geq \ell_{\min}^{\epsilon}(\mathcal{G}, S_1 \cup \tilde{S}_2)] \xrightarrow[m \to \infty]{} 0$ and together with $p_m \xrightarrow[m \to \infty]{} 0$, we have $\mathbb{E}_{S_1}[\mathrm{err}_P(h_{S_1})] \leq \delta_m^1 + \delta_{m,p,\alpha}^2 + o_m(1)$.

As a side note, computing the expectation over $S_1, S_2$ in the bound is impossible, due to the limited access of the training data. However, instead, we empirically estimate this term using a set of $k$ pairs $(S_1^i, S_2^i)$ of $m$ samples, yielding an additional term that scales as $\mathcal{O}(1/\sqrt{k})$ to the bound (see Prop. 2 in the appendix).

### 3.3 COMPARING PROP. 1 WITH STANDARD GENERALIZATION BOUNDS

Classic bounds (e.g., (Vapnik, 1998)) are based on bounding the test error with the sum between the train error together with a term $\mathcal{O}(\sqrt{\mathcal{C}(\mathcal{H})/m})$, where $\mathcal{C}(\mathcal{H})$ measures the complexity (e.g., VC dimension) of the class $\mathcal{H}$ (e.g., neural networks) and $m$ is the number of training samples. However, as discussed in Sec. 1, these bounds are vacuous in overparameterized learning regimes (e.g., training ResNet-50 on CIFAR10 classification). For instance, for VC-dimension based bounds (Vapnik, 1998), $\mathcal{C}(\mathcal{H})$ equals the VC-dimension of the class $\mathcal{H}$ which scales with the number of trainable parameters for ReLU networks (Bartlett et al., 2019). For example, even though the ResNet-50 architecture generalizes well when trained on CIFAR10, it has over 23 million parameters compared to the $m = 50000$ training samples in the dataset.

More recently, Neyshabur et al. (2015); Bartlett et al. (2017); Golowich et al. (2017); Neyshabur et al. (2018) suggested generalization bounds for neural networks that weakly depend on uniform convergence. In these bounds, the class-complexity $\mathcal{C}(\mathcal{H})$ is replaced with the individual complexity $\mathcal{C}(h_W)$ of the function we learn. For example, Golowich et al. (2017) proposed bounds that scale with $\mathcal{C}(h_W) = \rho^2 L$, where $L$ is the depth of $h_W$ and $\rho$ measures the product of the norms of its weight matrices. However, Nagarajan & Kolter (2019) showed that in certain cases unregularized least squares can generalize well even when its norm $\rho$ scales as $\Theta(\sqrt{m})$ and the bound becomes $\Theta_m(1)$. Furthermore, these bounds tend to be very large in practice (see Tab. 8 in (Neyshabur et al., 2019) and Tab. 1) and are negatively correlated with the test performance (Jiang et al., 2020). In addition, if the network's weight matrices' norms are larger than 1, quantities like $\rho$ grow exponentially when $L$ is varied. As shown in Tab. 1 this is empirically the case.

Our Prop. 1 offers a different way to measure generalization. Since this bound is not based on uniform convergence, it does not require that the network's complexity would be small in comparison to $m$; rather, the bound guarantees generalization if the network's effective size is smaller than that of a network that fits partially random labels. For instance, when the optimizer has a strong bias towards minimizing the effective depth, $\mathbb{E}_{\gamma}[\ell_{S_1}^{\epsilon}(h_{S_1}^{\gamma})] \approx \ell_{\min}^{\epsilon}(\mathcal{G}, S_1)$ which is by definition upper bounded by $\ell_{\min}^{\epsilon}(\mathcal{G}, S_1 \cup \tilde{S}_2)$. We note that $\ell_{\min}^{\epsilon}(\mathcal{G}, S_1 \cup \tilde{S}_2)$ grows to infinity as $m \to \infty$ (since the network needs to memorize $m \to \infty$ random labels). On the other hand, $\ell_{\min}^{\epsilon}(\mathcal{G}, S_1)$ is bounded by the depth of a network that approximates the target function $y$ up to an approximation error $\epsilon$ (which typically exists due to universal approximation arguments). Therefore, for sufficiently large $m$, we expect to have $\ell_{\min}^{\epsilon}(\mathcal{G}, S_1 \cup \tilde{S}_2) > \ell_{\min}^{\epsilon}(\mathcal{G}, S_1)$. As we empirically see in Sec. 4, the effective depths of SGD-trained networks are usually small.

Unlike previous bounds, our bound has the advantage of being fairly independent of $L$. Namely, when the minimal depth hypothesis (Obs. 1) holds, we expect $\mathbb{E}_{\gamma}[\ell_{S_1}^{\epsilon}(h_{S_1}^{\gamma})]$ to be unaffected by the depth $L$ of $h_{S_1}^{\gamma}$ (as long as $L \geq L_0$). Since $\ell_{\min}^{\epsilon}(\mathcal{G}, S_1 \cup \tilde{S}_2)$ is by definition independent of $L$, we expect $\mathbb{P}[\mathbb{E}_{\gamma}[\ell_{S_1}^{\epsilon}(h_{S_1}^{\gamma})] \geq \ell_{\min}^{\epsilon}(\mathcal{G}, S_1 \cup \tilde{S}_2)]$ to be independent of $L$ (when $L \geq L_0$). In Tab. 1 we empirically validate that our bound does not grow when increasing $L$.

## 4 EXPERIMENTS

In this section, we experimentally analyze the emergence of neural collapse in the intermediate layers of neural networks. First, we validate the "Minimal Depth Hypothesis" (Obs. 1). Following that, we look at how corrupted labels affect the extent of intermediate layer NCC separability and the $\epsilon$-effective depth. We show that as the number of corrupted labels in the data increases, so does

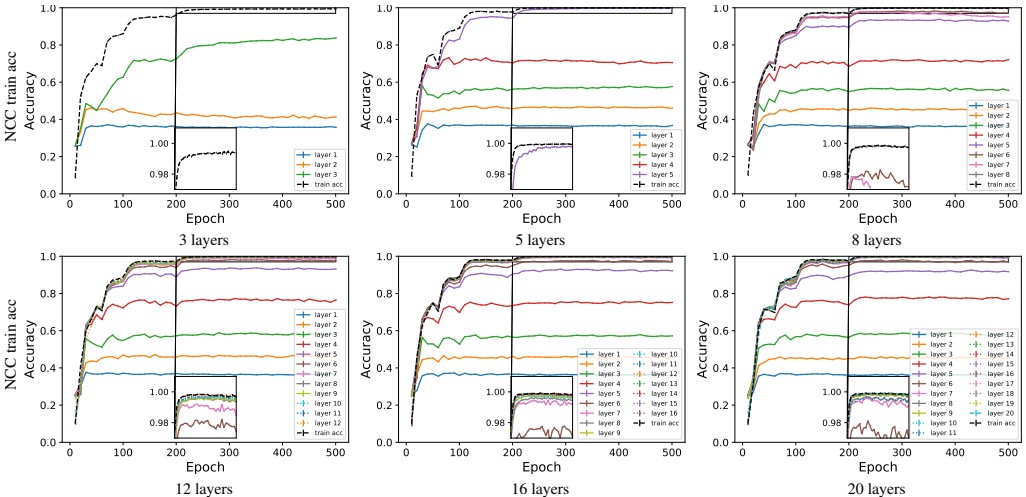

Figure 1: **Intermediate neural collapse of CONV-$L$-400 trained on CIFAR10.** We plot the NCC train accuracy rates of neural networks with varying numbers of hidden layers. Each curve stands for a different layer within the network.

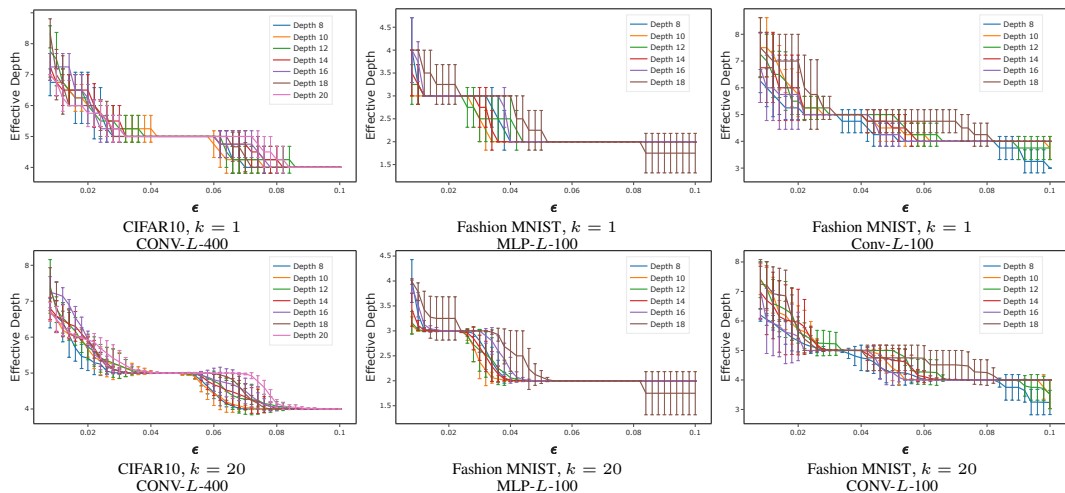

Figure 2: **Averaged $\epsilon$-effective depths over the last few epochs.** We plot the $\epsilon$-effective depth (y-axis) as a function of $\epsilon$ (x-axis). Each line specifies the $\epsilon$-effective depth of a neural network of a certain depth $L$. We show the averaged $\epsilon$-effective depth over the last $k = 1, 20$ epochs across 5 initializations. The network's architecture, dataset and $k$ are specified below each plot.

the $\epsilon$-effective depth. Finally, using the bound in Prop. 1, we provide non-trivial estimates of the test error. In Tab. 1, we empirically compare our bound with relevant baselines and show that, unlike other bounds, it achieves non-vacuous estimations of the test error. Throughout the experiments, we used Tesla-k80 GPUs for several hundred runs. Each run took between 5-20 hours. For additional experiments, see Appendix A. The plots are best viewed when zoomed in.

## 4.1 SETUP

**Training process.** We consider $k$-class classification problems (e.g., CIFAR10) and train multi-layered neural networks $h = e \circ f^L = e \circ g^L \circ \cdots \circ g^1 : \mathbb{R}^n \to \mathbb{R}^C$ on the corresponding training dataset $S$. The models are trained with SGD for cross-entropy loss minimization between its logits and the one-hot encodings of the labels. We consistently use batch size 128, learning rate schedule with an initial learning rate 0.1, decayed three times by a factor of 0.1 at epochs 60, 120, and 160, momentum 0.9 and weight decay $5e-4$. Each model is trained for 500 epochs.

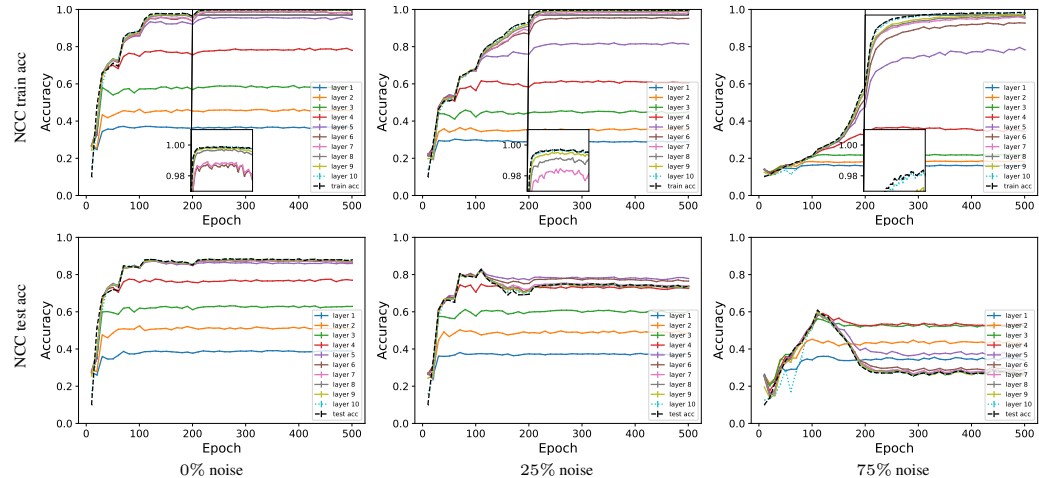

Figure 3: **Intermediate neural collapse of CONV-10-400 trained on CIFAR10 with partially corrupted labels.** We plot the NCC train/test accuracy rates of the various layers of a network trained with a certain amount of corrupted labels (see titles).

| Dataset | MNIST | | | Fashion MNIST | | | CIFAR10 | | | CIFAR10 |
|---|---|---|---|---|---|---|---|---|---|---|
| Architecture | CONV-10-50 | | | CONV-10-100 | | | CONV-16-100 | | | CONVRES-10-50 |
| $\mathbb{E}_{S_1, \gamma}[\mathrm{err}_P(h_{S_1}^{\gamma})]$ | 0.0075 | | | 0.0996 | | | 0.2676 | | | 0.29 |
| $p$ | 0.05 | 0.075 | 0.1 | 0.05 | 0.15 | 0.2 | 0.4 | 0.45 | 0.5 | 0.4 |
| Bound | 1.05 | 0.475 | **0.1** | 1.05 | 0.75 | **0.2** | **0.66** | 0.72 | 0.7 | **0.4** |

Table 2: Estimating the bound in Prop. 1. We used $\epsilon = 0.005$ to measure the effective depths.

**Architectures.** We used three types of architectures: (a) MLP-$L$-$H$ with $L$ fully-connected layers of width $H$, (b) CONV-$L$-$H$ with $L$ $3 \times 3$ convolutional layers with padding 1, stride 1 and $H$ output channels and (c) a residual convolutional network CONVRES-$L$-$H$ with $L$ residual blocks with two $3 \times 3$ convolutional layers. In each network the layers are interlaced with batch normalization layers and ReLU activations. For more details see Appendix A.1.

**Datasets.** We consider various datasets: MNIST, Fashion MNIST, and CIFAR10. For CIFAR10 we used random cropping, random horizontal flips, and random rotations (by $15k$ degrees for $k$ uniformly sampled from [24]). All datasets were standardized.

## 4.2 RESULTS

**Intermediate neural collapse.** To study the bias towards minimal depth, we trained a set of CONV-$L$-400 networks on CIFAR10 with varying depths. In each plot at Fig. 1 we report the train NCC classification accuracy rates for each intermediate layer of a network of a certain depth. We make multiple interesting observations; **(i)** For networks with 8 or higher hidden layers, the eighth and higher layers exhibit NCC train accuracy of approximately $100\%$, and therefore, are effectively of depth 7. **(ii)** We observe that neural collapse strengthens when increasing the network's depth, on both train and test data. **(iii)** The embeddings of the top layers become NCC separable approximately at the same epoch. **(iv)** The degree of NCC separability of intermediate layer $i$ converges as a function of $L$. The results of this experiment are substantially extended and repeated with different architectures and datasets in the appendix (see Figs. 5-13). In these experiments we report the NCC train and test accuracy rates along with additional measures of neural collapse when varying the depth. Specifically, in Figs. 4 and 5 we report the results with CONVRES-$L$-500.

**The effect of the depth on the $\epsilon$-effective depth.** In Obs. 1 we claimed that the $\epsilon$-effective depth is insensitive to the actual depth of the network (once it exceeds a certain threshold). To validate this hypothesis we conducted the following experiments. We trained models on MNIST, Fashion MNIST and CIFAR10 with varying depth $L$. In Fig. 2 we plotted the averaged $\epsilon$-effective depths of each network's last $k = 1, 20$ epochs as a function of $\epsilon$. We also average the results across 5

different weight initializations and plot them along with error bar standard deviations. As can be seen, the $\epsilon$-effective depth is almost unaffected by the choice of $L$ for a given $\epsilon$. Remarkably, for each $\epsilon$, the averaged effective depth varies very little across the various networks. Differently said, the $\epsilon$-effective depths of two trained deep networks of different depths are more or less the same, validating our Minimal Depth Hypothesis.

**NCC separability with partially corrupted labels.** Simply put, Prop. 1 compares the depths required to fit correct labels and partially corrupt labels. To better understand the effect of corrupted labels on the complexity of the task, we compare the $\epsilon$-effective depths of models trained with varying amounts of corrupted labels. Namely, we study the *degree* of NCC separability in the intermediate layers of neural networks that are trained with varying amounts of corrupted labels.

For this experiment we trained instances of CONV-10-400 for CIFAR10 classification with $0\%$, $10\%$ and $75\%$ corrupted labels (e.g., uniformly distributed random labels). We plot the degrees of NCC separation on the train and test sets, $1 - \text{err}_S(\hat{h}_i)$ and $1 - \text{err}_P(\hat{h}_i)$, across the intermediate layers of the neural networks during the optimization procedure.

As can be seen in Fig. 3, when increasing the amount of random labels, the degree of NCC separability across the intermediate layers tend to decrease. For example, when training with $\geq 25\%$ corrupted labels, the sixth layer's NCC accuracy rate drops lower than $98\%$, in comparison with training without corrupted labels that gives us $> 98\%$ accuracy. In particular, the $\epsilon$-effective depth of the former network is 6 while the latter's is 5, when $\epsilon = 0.02$ (see Def. 1). This experiment is extended and repeated in a variety of settings in Figs. 14-18.

**Estimating the bound in equation 3.** We estimate the bound in equation 3 for multiple architectures and datasets. In each case we used $\epsilon = 0.005$ by default and employed different 'guesses' $p$ (see Tab. 2) depending on the complexity of the learning task. We report an estimation of the expected test error of the models, $\mathbb{E}_{S_1,\gamma}[\text{err}_P(h_{S_1}^\gamma)]$ and an estimation of the bound for each selection of $p$. For concrete technical details, see Appendix A.

As can be seen, for appropriate selections of $p$, we obtained non-trivial estimates to the test performance of the models, which are almost unheard of when it comes to standard bounds for deep neural networks. As expected, if the guess $p$ is overoptimistic (e.g., close to $\mathbb{E}_{S_1,\gamma}[\text{err}_P(h_{S_1}^\gamma)]$), then, the first term in the bound tends to be large compared to $\mathbb{E}_{S_1,\gamma}[\text{err}_P(h_{S_1}^\gamma)]$.

**Comparing our bound with standard generalization bounds.** Since the $\epsilon$-effective depth of sufficiently deep neural networks is insensitive to depth (see Fig. 2), we expect the bound to be insensitive to depth as well. We estimate the bound in equation 3 for CONV-$L$-50 trained on MNIST and CONV-$L$-100 trained on Fashion MNIST and CIFAR10 with $L = 10, 12, 15$ for the first two and with $L = 15, 18, 20$ for CIFAR10. As shown in Tab. 1, we obtain similar bounds for each depth. Finally, we compare our bound to several baseline generalization bounds for deep networks to show that it outperforms traditional generalization bounds. We used the implementation of Neyshabur et al. (2019) to compute the bounds. **While our bound is empirically non-vacuous and fairly independent of depth, the traditional bounds are extremely vacuous and rapidly grow when increasing the depth.**

## 5 CONCLUSIONS

Understanding the ability of SGD to generalize well when training overparameterized neural network is attributed as one of the major open problems in deep learning theory (Zhang et al., 2017). In this paper we offer a new angle to study the role of depth in deep learning and the connection between neural collapse and generalization.

We characterize a notion of effective depth that measures the lowest layer that enjoys NCC separability. We introduce a novel generalization bound that measures the likelihood in which the effective depth of a trained neural network is (strictly) smaller than the minimal depth required to achieve NCC separability with partially corrupted labels. This criterion, as demonstrated empirically, is a good predictor of generalization. Furthermore, we characterize and empirically demonstrate that when sufficiently deep networks are trained, they converge to the same effective depth, implying that our bound is fairly constant when the depth is varied.

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

# A ADDITIONAL EXPERIMENTS AND DETAILS

## A.1 ARCHITECTURES

In this section we give a detailed description of the architectures used in the experiments.

The first architecture is a convolutional network, denoted by CONV-$L$-$H$. The network starts with a stack of a $2 \times 2$ convolutional layer with stride 2, batch normalization, a convolution of the same structure, batch normalization, and ReLU. Following that we have a set of $L$ stacks of blocks $g^i(x) = \sigma(B_i(C_i(x)))$, where $C_i$ is a $3 \times 3$ convolutional layers with $H$ channels, stride 1 and padding 1, $B_i$ is a batch normalization layer, and $\sigma$ is the ReLU activation. The last layer is linear. When computing the effective depth, the $i$'th intermediate layer refers to the output of the $i$'th block of $g^i$.

The second architecture is an MLP, denoted by MLP-$L$-$H$ consisting of $L$ hidden layers, where each layer $g^i(x) = \sigma(B_i(T_i(x)))$ contains a linear layer $T_i$ of output width $H$, followed by batch normalization $B_i$ and ReLU activation $\sigma$. The last layer is linear.

The third architecture is a convolutional residual network denoted by CONVRES-$L$-$H$. The network starts with a stack of a $2 \times 2$ convolutional layer with stride 2, batch normalization, a convolution of the same structure, batch normalization, and ReLU. Following that we have a set of $L$ residual blocks, where each block computes $g^i(x) = \sigma(x + B_i^2(C_i^2(\sigma(B_i^1(C_i^1(x))))))$, where each $C_i^j$ is a $3 \times 3$ convolutional layer with $H$ channels, stride 1 and padding 1, $B_i^j$ is a batch normalization layer and $\sigma$ is the ReLU activation. The last layer is linear.

## A.2 ESTIMATING THE GENERALIZATION BOUND

In this section we describe how we empirically estimate the bound in Prop. 1.

**Estimating the bound.** We would like to estimate the first term in the bound,

$$\mathbb{P}_{S_1, S_2, \tilde{Y}_2} \left[ \mathbb{E}_\gamma [\ell^\epsilon_{S_1}(h^\gamma_{S_1})] \geq \ell^\epsilon_{\min}(\mathcal{G}, S_1 \cup \tilde{S}_2) \right]. \tag{4}$$

According to Prop. 2 in order to estimate this term we need to generate i.i.d. triplets $(S_1^i, S_2^i, \tilde{Y}_2^i)$. Since we have a limixted access to training data, we use a variation of cross-validation and generate $k_1 = 5$ i.i.d. disjoint splits $(S_1^i, S_2^i)$ of the training data $S$. For each one of these pairs, we generate $k_2 = 3$ corrupted labelings $\tilde{Y}_2^{ij}$. We denote by $\tilde{S}_2^{ij}$ the set obtained by replacing the labels of $S_2^i$ with $\tilde{Y}_2^{ij}$ and $\tilde{S}_3^{ij} := S_1^i \cup \tilde{S}_2^{ij}$.

As a first step, we would like to estimate $\mathbb{E}_\gamma [\ell^\epsilon_{S_1^i}(h^\gamma_{S_1^i})]$ for each $i \in [k_1]$. For this purpose, we randomly select $T_1 = 5$ different initializations $\gamma_1, \dots, \gamma_{T_1}$ and for each one, we train the model $h^{\gamma_t}_{S_1^i}$ using the training protocol described in Sec. 4.1. Once trained, we compute $\ell^\epsilon_{S_1}(h^{\gamma_t}_{S_1^i})$ for each $t \in [T_1]$ (see Def. 1) and approximate $\mathbb{E}_\gamma [\ell^\epsilon_{S_1^i}(h^\gamma_{S_1^i})]$ using $d_i := \frac{1}{T_1} \sum_{t=1}^{T_1} \ell^\epsilon_{S_1^i}(h^{\gamma_t}_{S_1^i})$.

As a next step, we would like to evaluate $\mathbb{I}[d_i \geq \ell^\epsilon_{\min}(\mathcal{G}, \tilde{S}_3^{ij})]$. We notice that $d_i \geq \ell^\epsilon_{\min}(\mathcal{G}, S_1^i \cup \tilde{S}_2^i)$ if and only if there is a $d_i$-layered neural network $f = g^{d_i} \circ \cdots \circ g^1$ for which $\text{err}_{\tilde{S}_3^{ij}}(\hat{h}) \leq \epsilon$, where $\hat{h}(x) := \arg\min_{c \in [C]} \|f(x) - \mu_f(S_c)\|$. In general, computing this Boolean value is computationally hard. Therefore, to estimate this Boolean value, we simply train a $(d_i + 1)$-layered network $h = e \circ f$ and check whether its penultimate layer is $\epsilon$-NCC separable, i.e., $\text{err}_{\tilde{S}_3^{ij}}(\hat{h}) \leq \epsilon$, where $\hat{h}(x) := \arg\min_{c \in [C]} \|f(x) - \mu_f(S_c)\|$. If SGD implicitly optimizes neural networks to maximize NCC separability as observed in (Papyan et al., 2020) (and also in this paper), we should expect to obtain $\epsilon$-NCC separability in the penultimate layer if that is possible with a $d_i$-layered network. Since training might be non-optimal, to obtain a robust estimation, we train $T_2 = 5$ models $h_t = e_t \circ f_t$ of depth $d_i + 1$ and pick the one with the best NCC separability in its penultimate layer. Namely, we replace $\ell^\epsilon_{\min}(\mathcal{G}, \tilde{S}_3^{ij})$ with $\min_{t \in [T_2]} \ell^\epsilon_{\tilde{S}_3^{ij}}(h_t)$ and estimate $\mathbb{I}[d_i \geq \ell^\epsilon_{\min}(\mathcal{G}, \tilde{S}_3^{ij})]$ using $\mathbb{I}[d_i \geq \min_{t \in [T_2]} \ell^\epsilon_{\tilde{S}_3^{ij}}(h_t)]$.

Our final estimation is the following

$$\frac{1}{k_1}\sum_{i=1}^{k_1}\frac{1}{k_2}\sum_{j=1}^{k_2}\mathbb{I}\left[d_i \geq \min_{t\in[T_2]}\mathscr{d}_{\tilde{S}_3^{ij}}^{\epsilon}(h_t)\right] \approx \mathbb{P}_{S_1,S_2,\tilde{Y}_2}\left[\mathbb{E}_{\gamma}[\mathscr{d}_{S_1}^{\epsilon}(h_{S_1}^{\gamma})] \geq \mathscr{d}_{\min}^{\epsilon}(\mathcal{G},S_1\cup\tilde{S}_2)\right]. \quad (5)$$

In order to estimate the bound we assume that $\delta_m^1$ and $\delta_{m,p,\alpha}^2$ are negligible constants and that $\alpha = 1$. The estimation of the bound is given by the sum of the left hand side in equation 5 and $p$.

**Estimating the mean test error.** To estimate the mean test error, $\mathbb{E}_{S_1,\gamma}[\text{err}_P(h_{S_1}^{\gamma})]$, as typically done in machine learning, we replace the population distribution $P$ with the test set $S_{test}$ and we replace the expectation over $S_1$ and $\gamma$ with averages across the $k_1 = 5$ random selections of $\{S_1^i\}_{i=1}^{k_1}$ and $T_1 = 5$ random selections of $\{\gamma_t\}_{t=1}^{T_1}$. Namely, we compute the following $\frac{1}{k_1}\sum_{i=1}^{k_1}\frac{1}{T_1}\sum_{t=1}^{T_1}\text{err}_{S_{test}}(h_{S_1^i}^{\gamma_t}) \approx \mathbb{E}_{S_1,\gamma}[\text{err}_P(h_{S_1}^{\gamma})]$.

### A.3 NEURAL COLLAPSE

To obtain a comprehensive analysis of collapse across layers, we also estimate the degree of NC1.

To evaluate NC1, we follow the process suggested by Galanti et al. (2022a), which is a simplified version of the original approach of Papyan et al. (2020). For a feature map $f : \mathbb{R}^d \to \mathbb{R}^p$ and two (class-conditional) distributions[1] $Q_1, Q_2$ over $\mathcal{X} \subset \mathbb{R}^d$, we define their *class-distance normalized variance* (CDNV) to be

$$V_f(Q_1,Q_2) := \frac{\text{Var}_f(Q_1) + \text{Var}_f(Q_2)}{2\|\mu_f(Q_1) - \mu_f(Q_2)\|^2},$$

where $\mu_u(Q) := \mathbb{E}_{x\sim Q}[u(x)]$ and by $\text{Var}_u(Q) := \mathbb{E}_{x\sim Q}[\|u(x) - \mu_u(Q)\|^2]$ the mean and variance of $u(x)$ for $x \sim Q$. Essentially, this quantity measures to what extent the feature vectors of samples from $Q_1$ and $Q_2$ are separated and clustered in space.

To demonstrate the gradual evolution of collapse across the layers, for each sub-architecture $f^i = g^i \circ \cdots \circ g^1(x)$ we consider the train and test class features variations $\text{Avg}_{c\neq c'}[V_{f^i}(S_c, S_{c'})]$ and $\text{Avg}_{c\neq c'}[V_{f^i}(P_c, P_{c'})]$. The population distribution of each class, $P_c$, is replaced with the test samples of that class.

As shown by Galanti et al. (2022a), this definition is essentially the same as that of Papyan et al. (2020). Furthermore, they showed that the NCC classification error rate can be upper bounded in terms of the CDNV. However, the NCC error can be zero in cases where the CDNV is larger than zero. For example, if the two classes are uniformly distributed over the 1-radius circles around the points $(-1, 0)$ and $(1, 0)$ in $\mathbb{R}^2$, then they are perfectly NCC separable while the CDNV between the two distributions is 0.25.

**Auxiliary experiments on the effective depth.** In Figs. 7-13 we plot the CDNV and the NCC accuracy rates of neural networks with varying numbers of hidden layers evaluated on the train and test data. Each curve stands for a different layer within the network. As can be seen, in all cases, for networks deeper than a threshold we obtain (near perfect) NCC separability in all of the top layers. Furthermore, the degree of neural collapse seems to improve with the network's depth.

**Auxiliary experiments with noisy labels.** In Figs. 14-18 we repeat the experiment in Fig. 3 and plot the results of the same experiment, with different networks and datasets (see captions). As can be seen, the effective NCC depth of a neural network tends to increase as we train with increasing amounts of corrupted labels.

---

[1]The definition can be extended to finite sets $S_1, S_2 \subset \mathcal{X}$ by defining $V_f(S_1, S_2) = V_f(U[S_1], U[S_2])$.

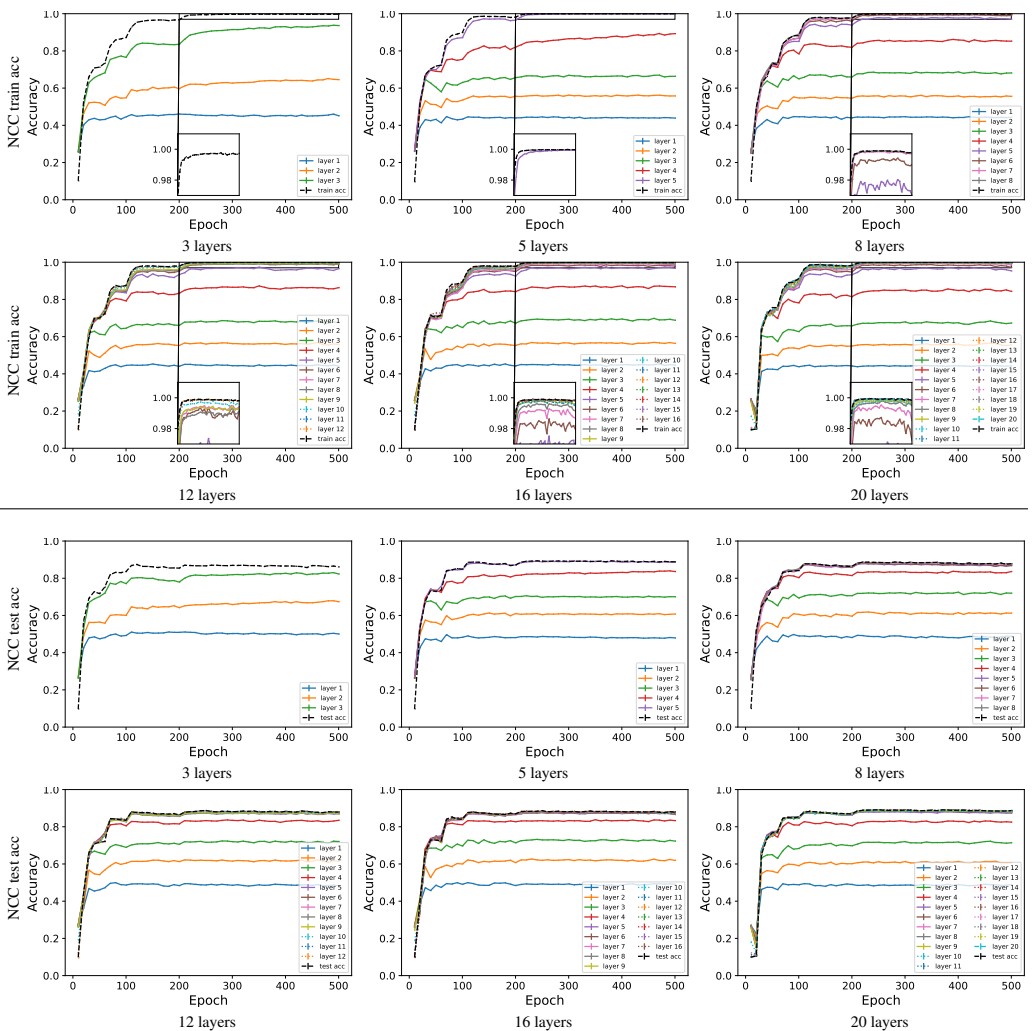

Figure 4: **Intermediate neural collapse of CONVRES-$L$-500 trained on CIFAR10.** We plot the NCC train and test accuracy rates of neural networks with varying numbers of hidden layers. Each curve stands for a different layer within the network.

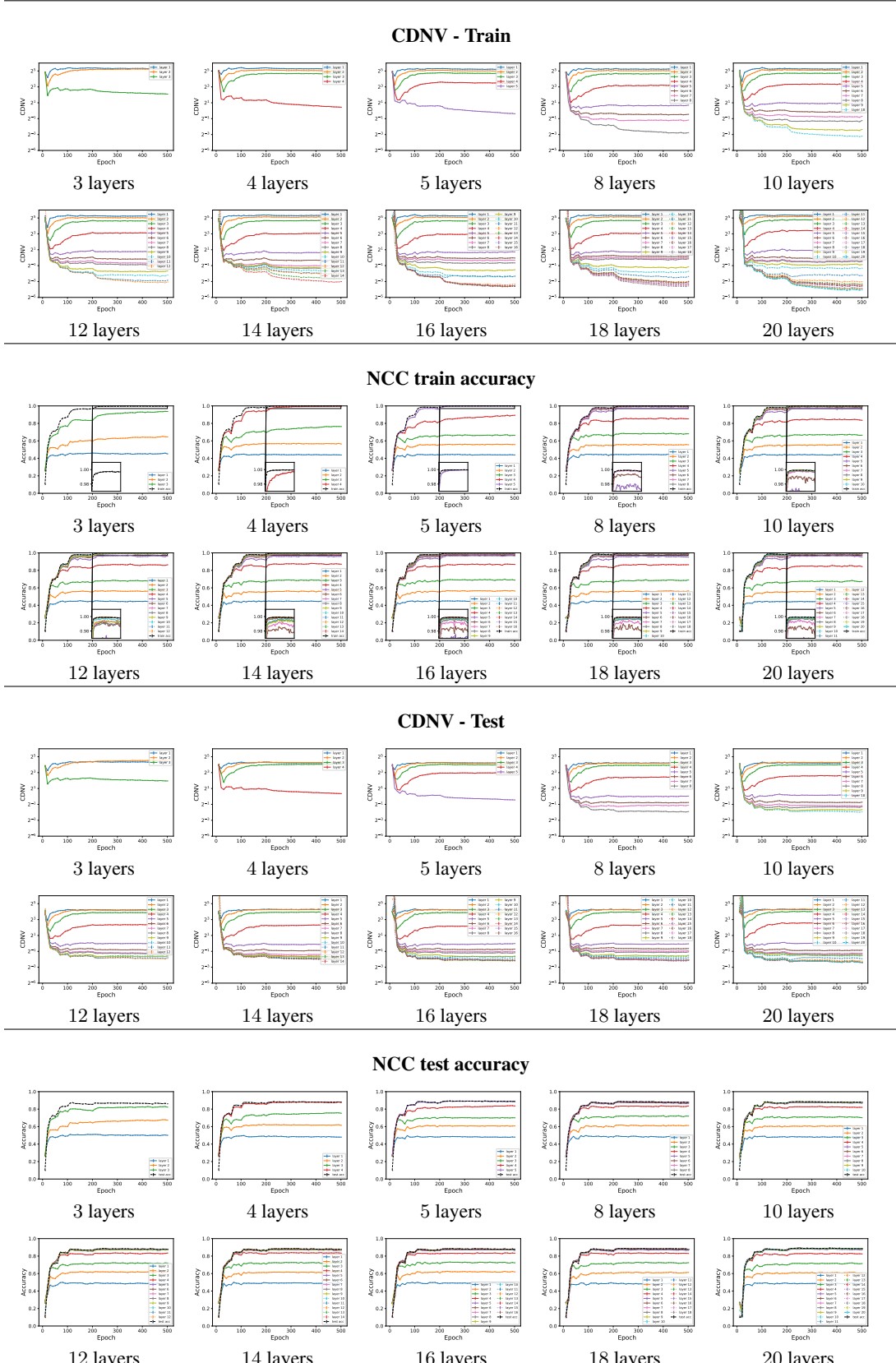

Figure 5: **Intermediate neural collapse of CONVRES-$L$-500 trained on CIFAR10.** See Fig. 1 in the main text for details.

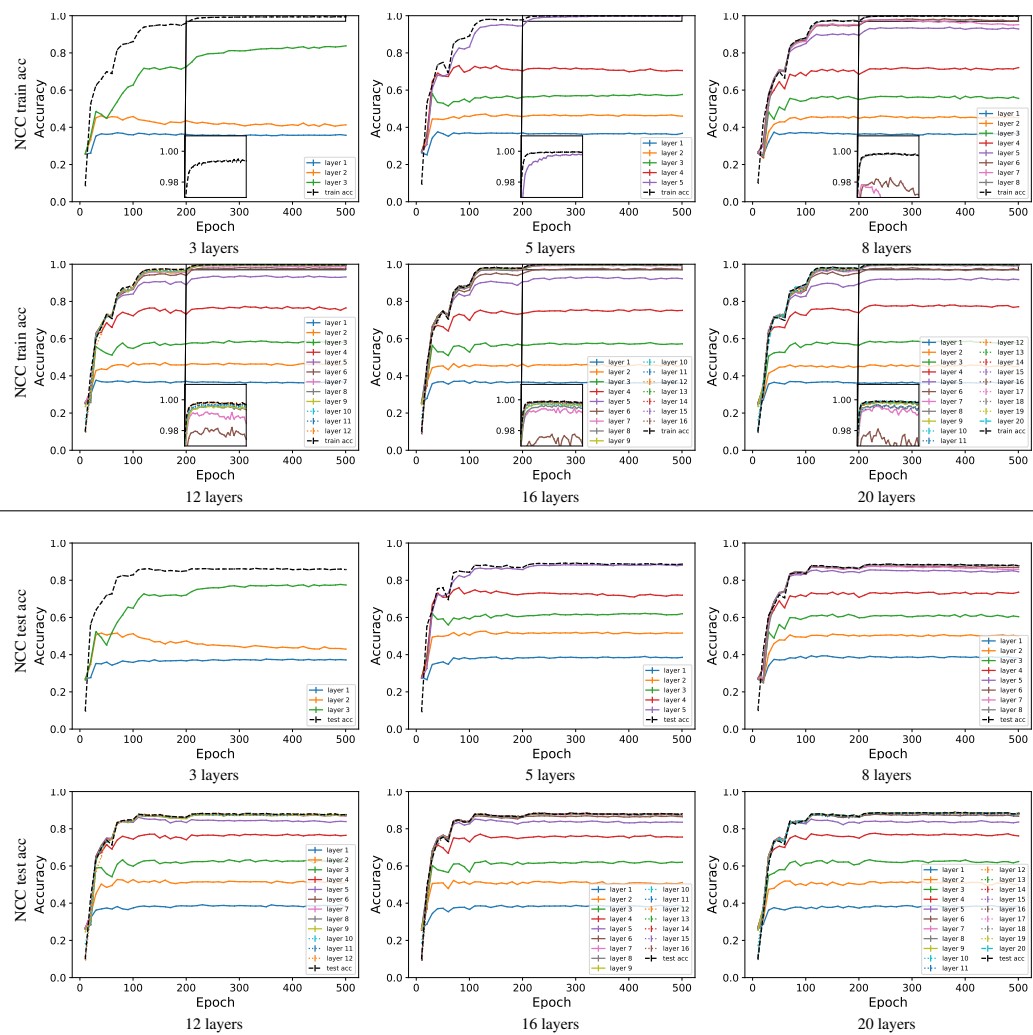

Figure 6: **Intermediate neural collapse of CONV-$L$-400 trained on CIFAR10.** We plot the NCC train and test accuracy rates of neural networks with varying numbers of hidden layers. Each curve stands for a different layer within the network.

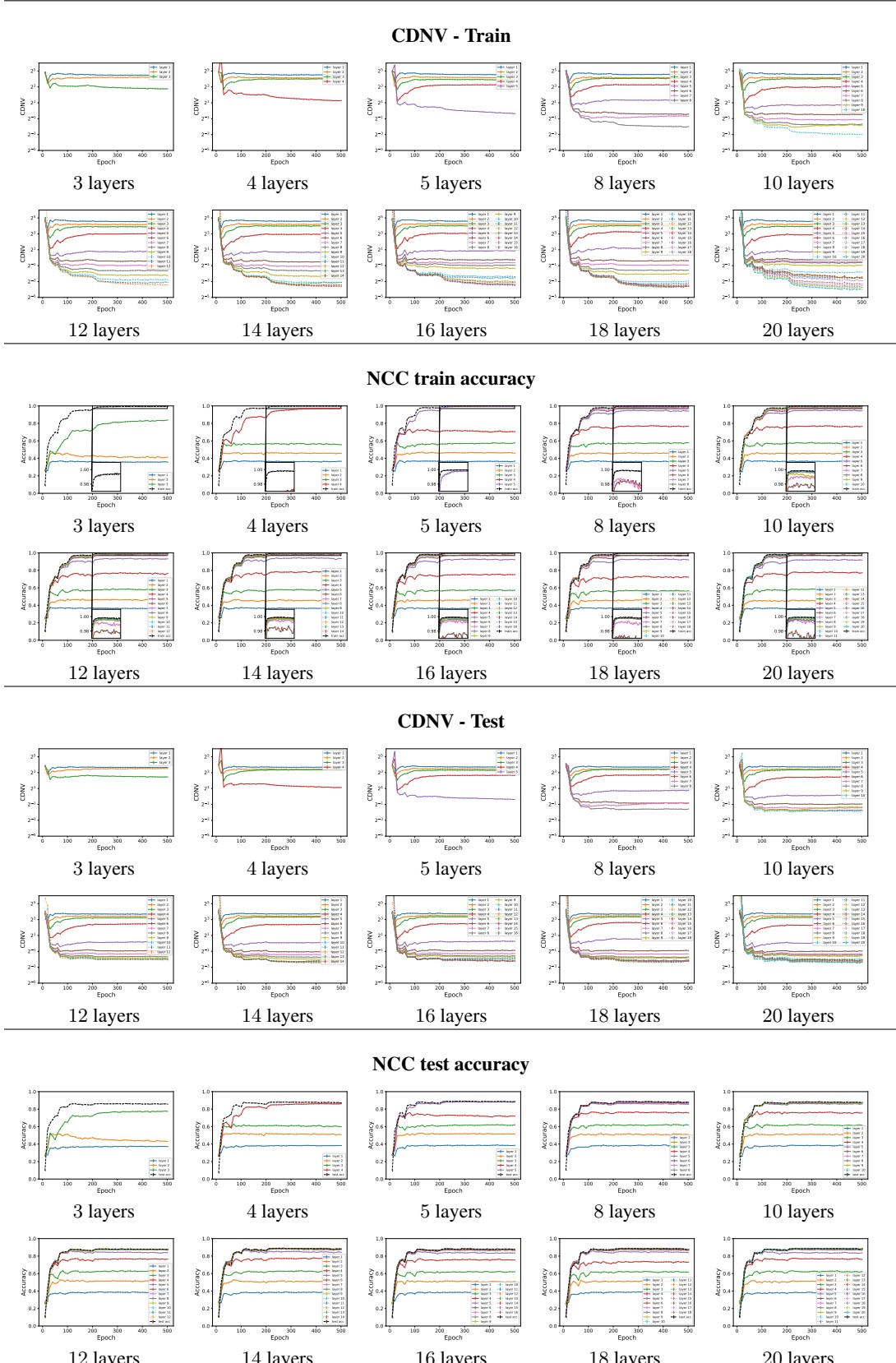

Figure 7: **Intermediate neural collapse of CONV-$L$-400 trained on CIFAR10.** See Fig. 1 in the main text for details.

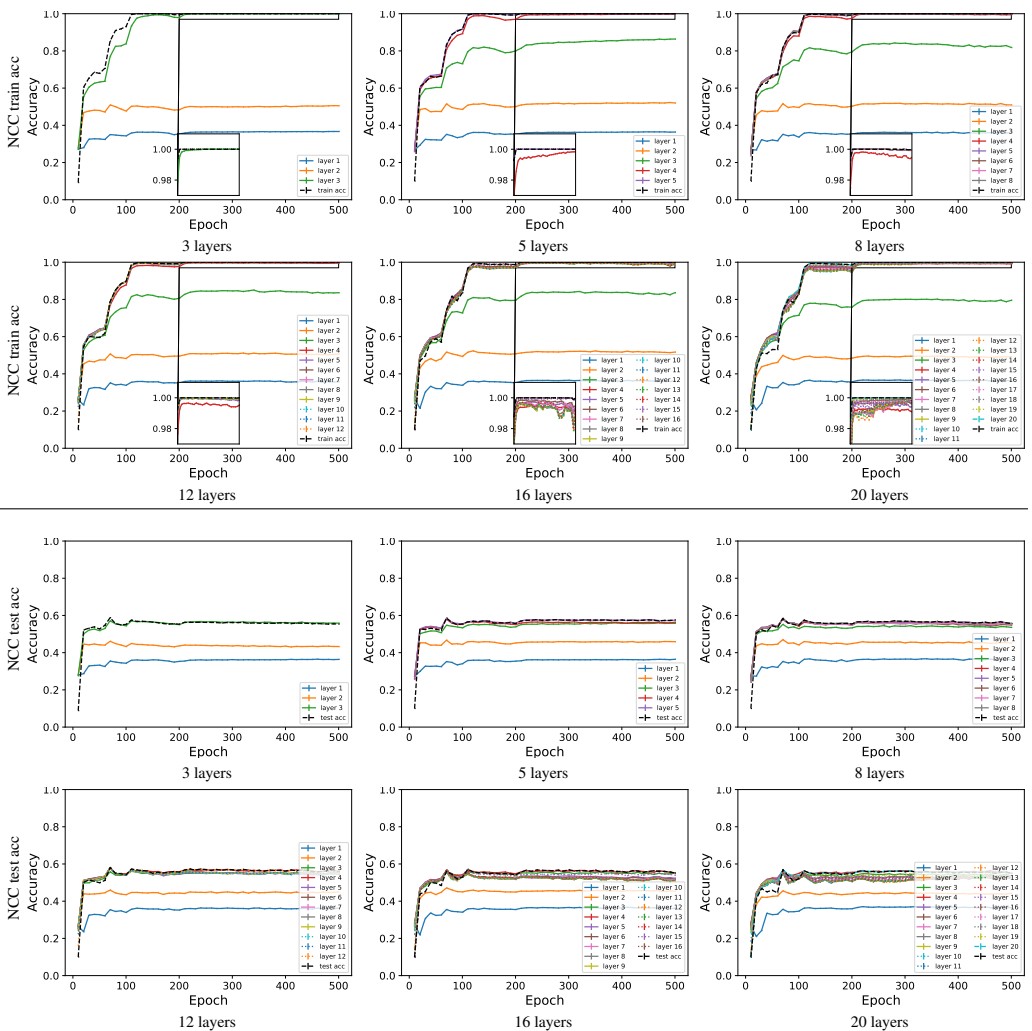

Figure 8: **Intermediate neural collapse of MLP-$L$-300 trained on CIFAR10.** We plot the NCC train and test accuracy rates of neural networks with varying numbers of hidden layers. Each curve stands for a different layer within the network.

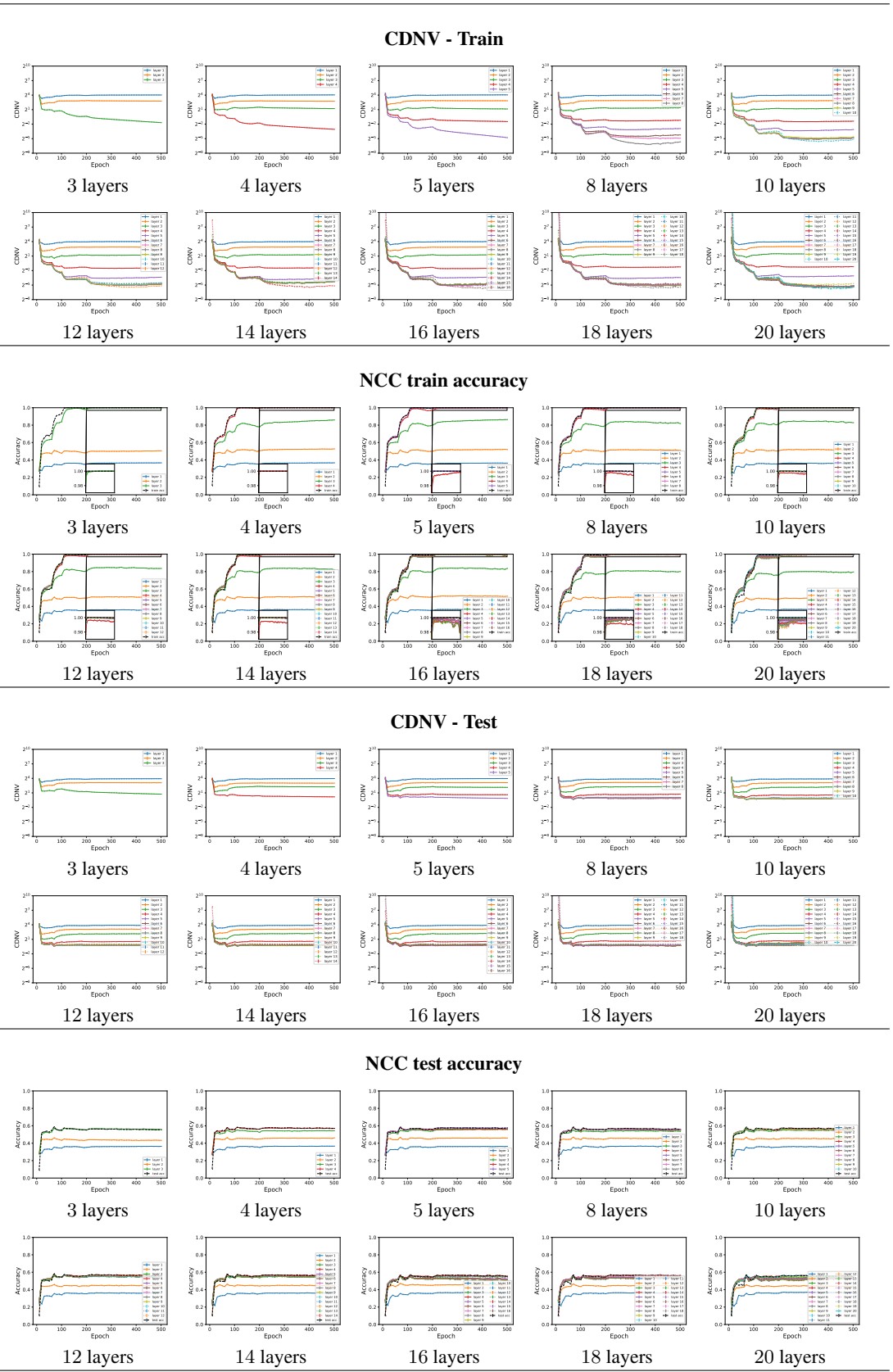

Figure 9: **Intermediate neural collapse of MLP-$L$-300 trained on CIFAR10.** See Fig. 1 in the main text for details.

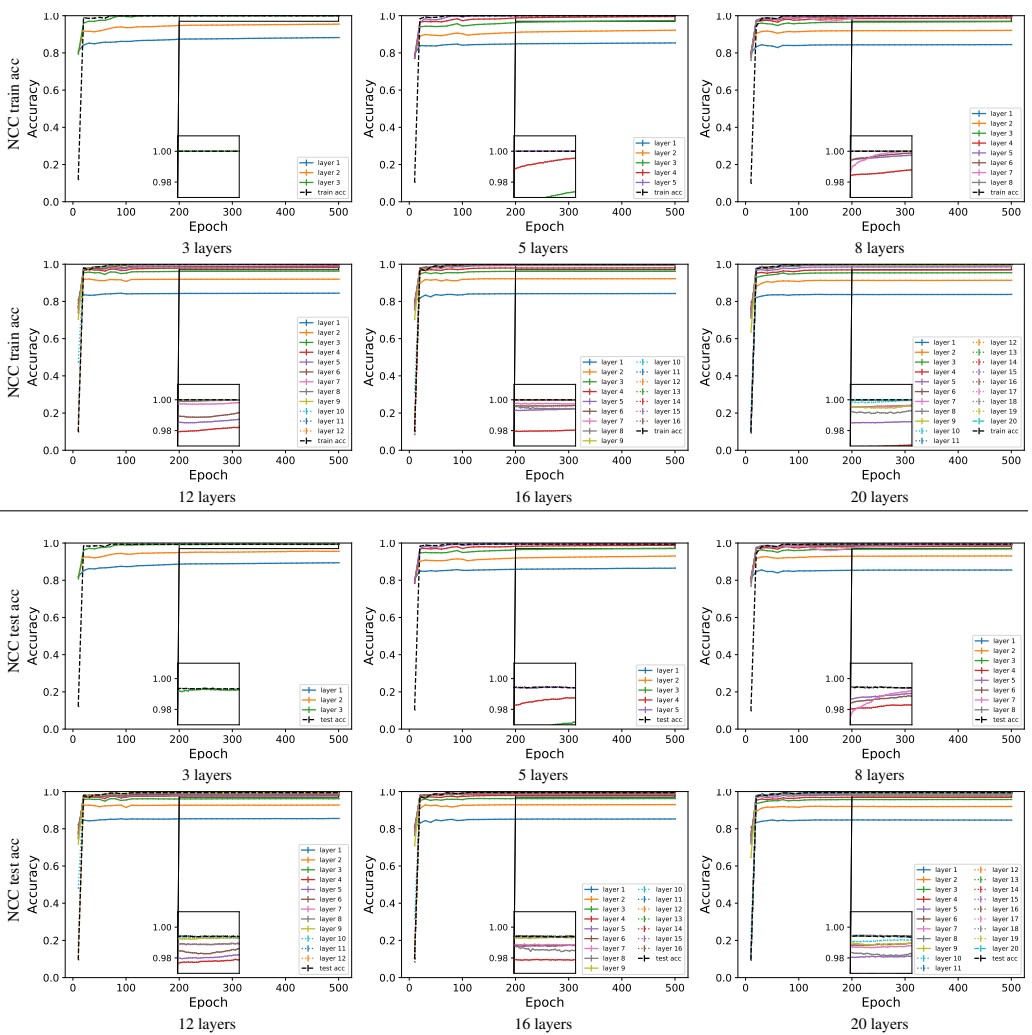

Figure 10: **Intermediate neural collapse of CONV-$L$-50 trained on MNIST.** We plot the NCC train and test accuracy rates of neural networks with varying numbers of hidden layers. Each curve stands for a different layer within the network.

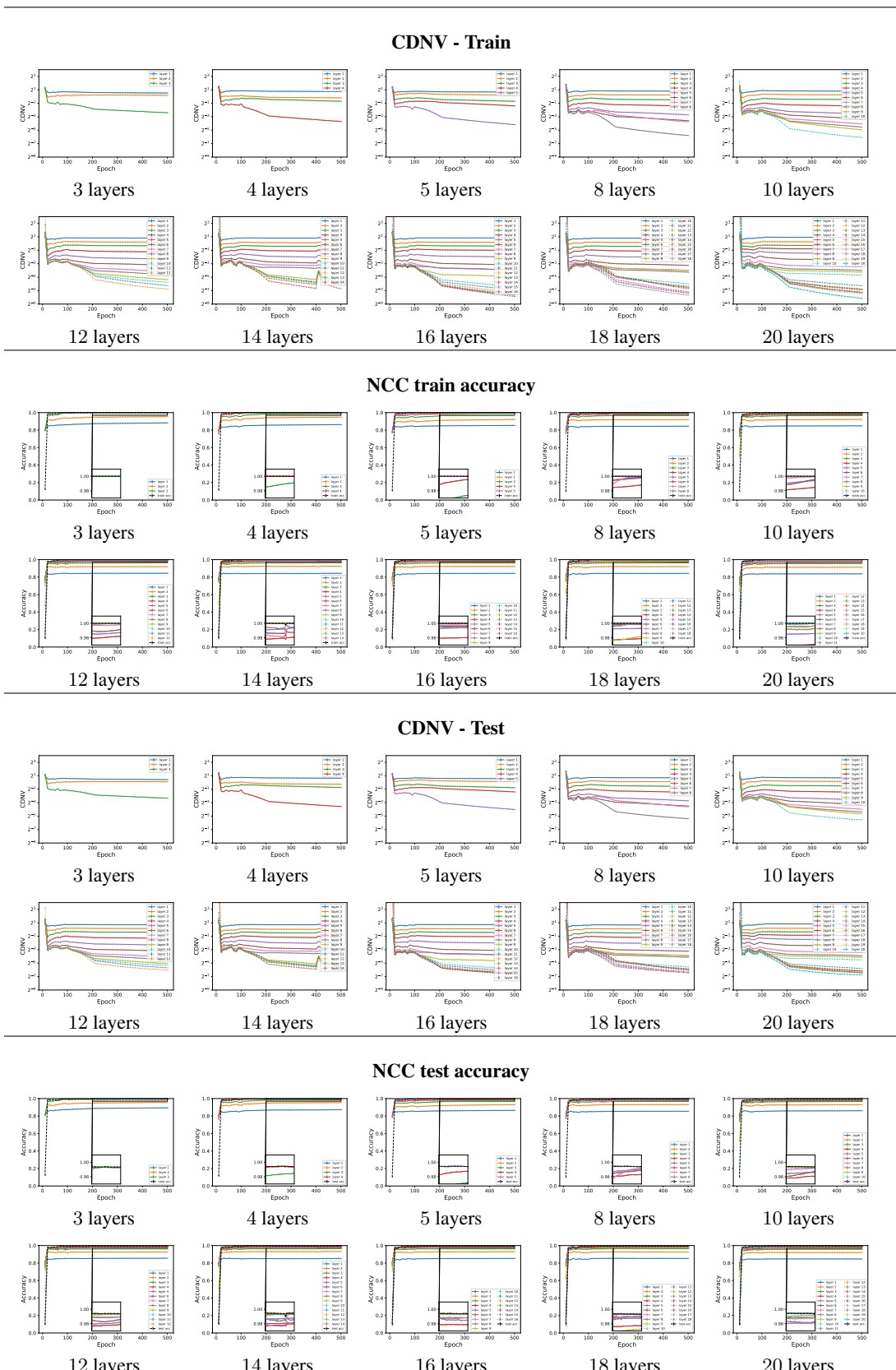

Figure 11: **Intermediate neural collapse of CONV-$L$-50 trained on MNIST.** See Fig. 1 in the main text for details.

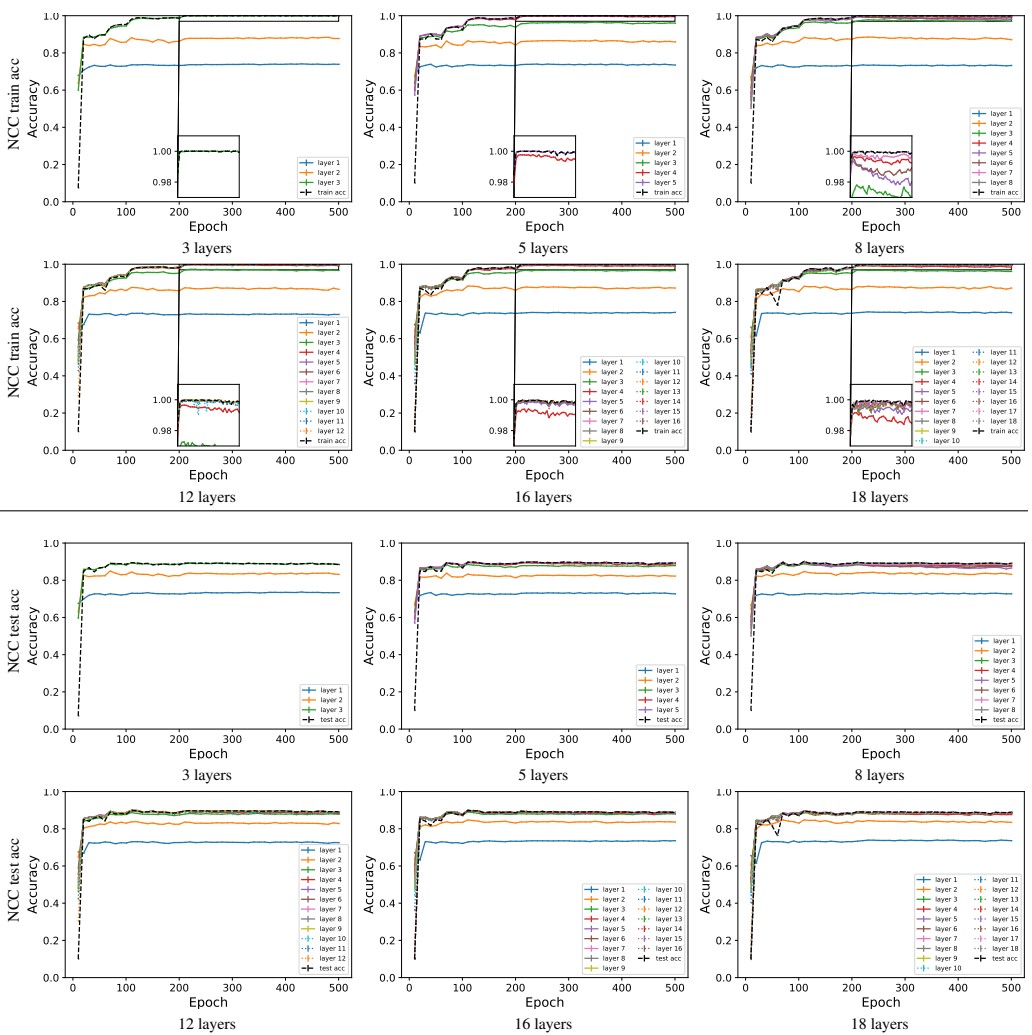

Figure 12: **Intermediate neural collapse of MLP-$L$-100 trained on Fashion MNIST.** We plot the NCC train and test accuracy rates of neural networks with varying numbers of hidden layers. Each curve stands for a different layer within the network.

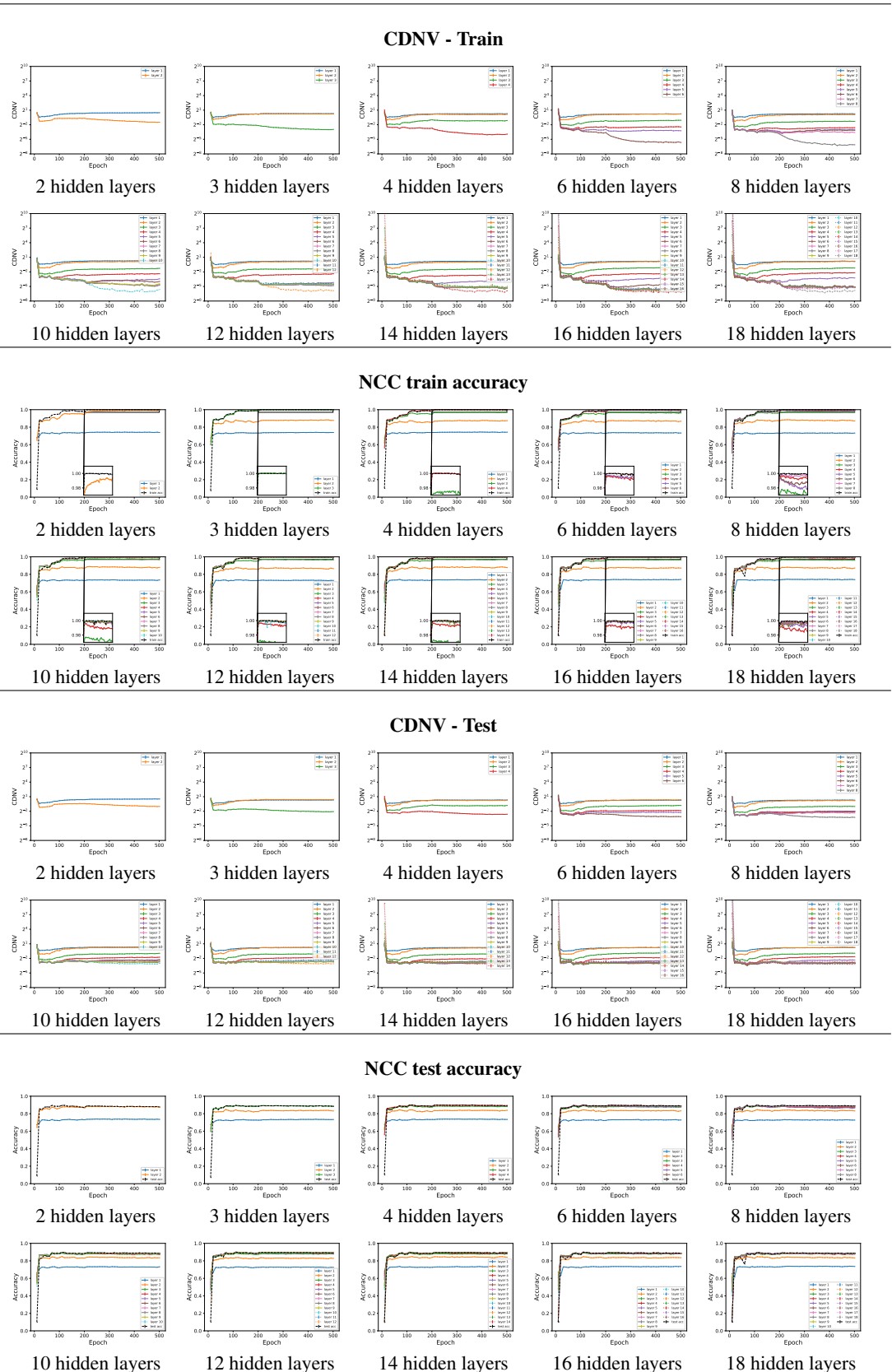

Figure 13: **Intermediate neural collapse of MLP-$L$-100 trained on Fashion MNIST.** See Fig. 1 in the main text for details.

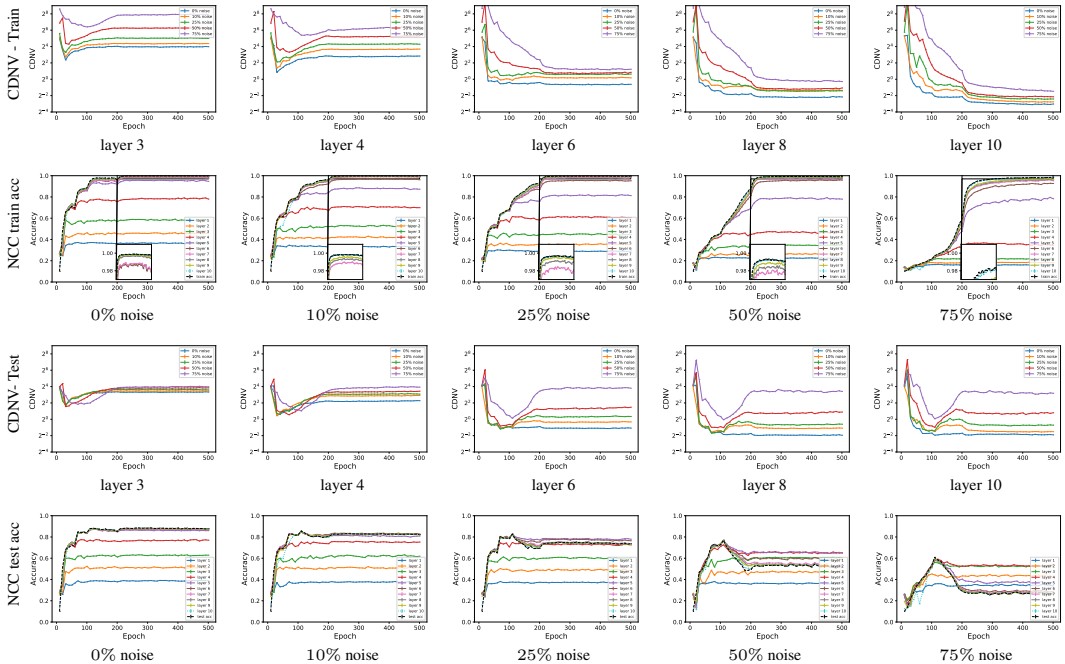

Figure 14: **Intermediate neural collapse of CONV-10-400 trained on CIFAR10 with partially corrupted labels.** In the first (third) row, we plot the CDNV on the train (test) data for intermediate layers of networks trained with varying amounts of corrupted labels (see legend). In the second (fourth) row, we plot the NCC accuracy rates of the various layers of a network trained with a certain amount of corrupted labels (see titles).

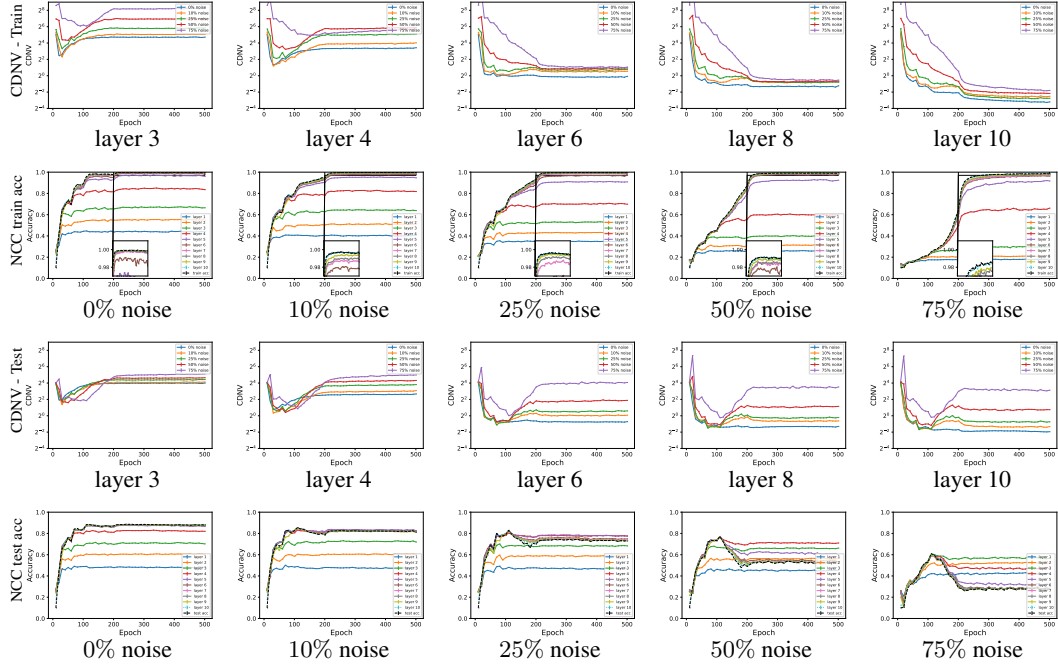

Figure 15: **Intermediate neural collapse of CONVRES-10-500 trained on CIFAR10 with noisy labels.** See Fig 3 in the main text for details.

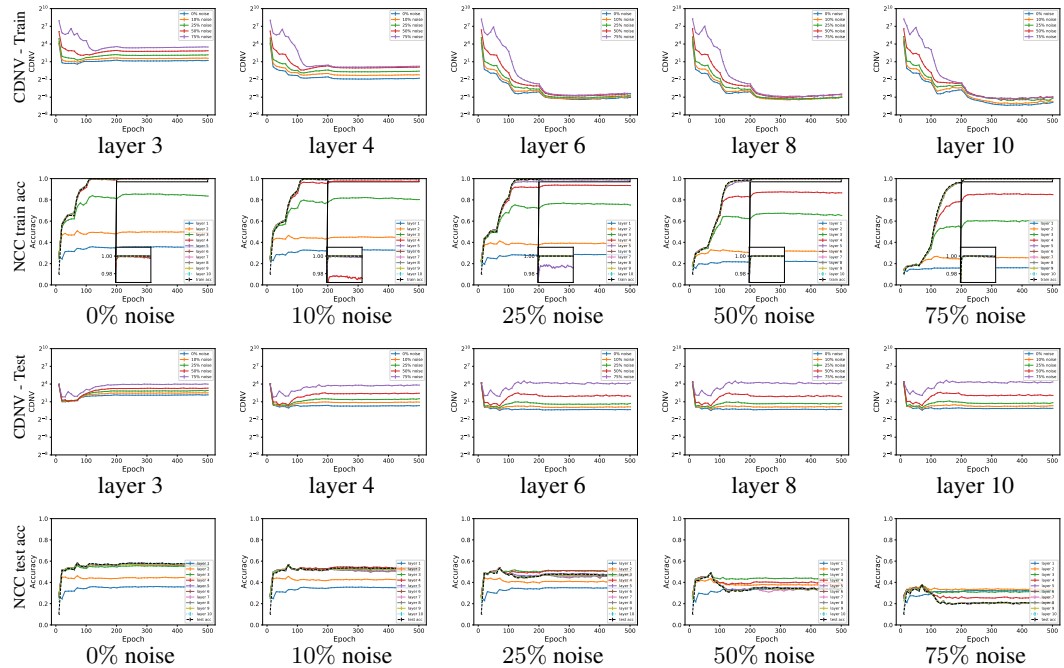

Figure 16: **Intermediate neural collapse of MLP-10-500 trained on CIFAR10 with noisy labels.** See Fig 3 in the main text for details.

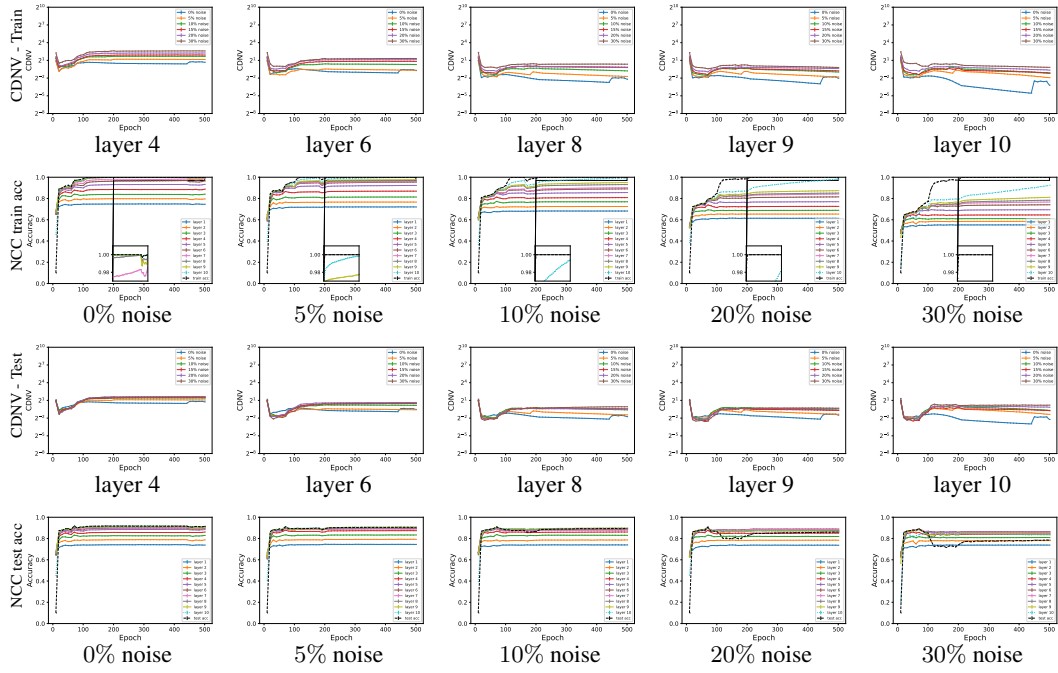

Figure 17: **Intermediate neural collapse of CONV-10-100 trained on Fashion MNIST with noisy labels.** See Fig. 3 in the main text for details.

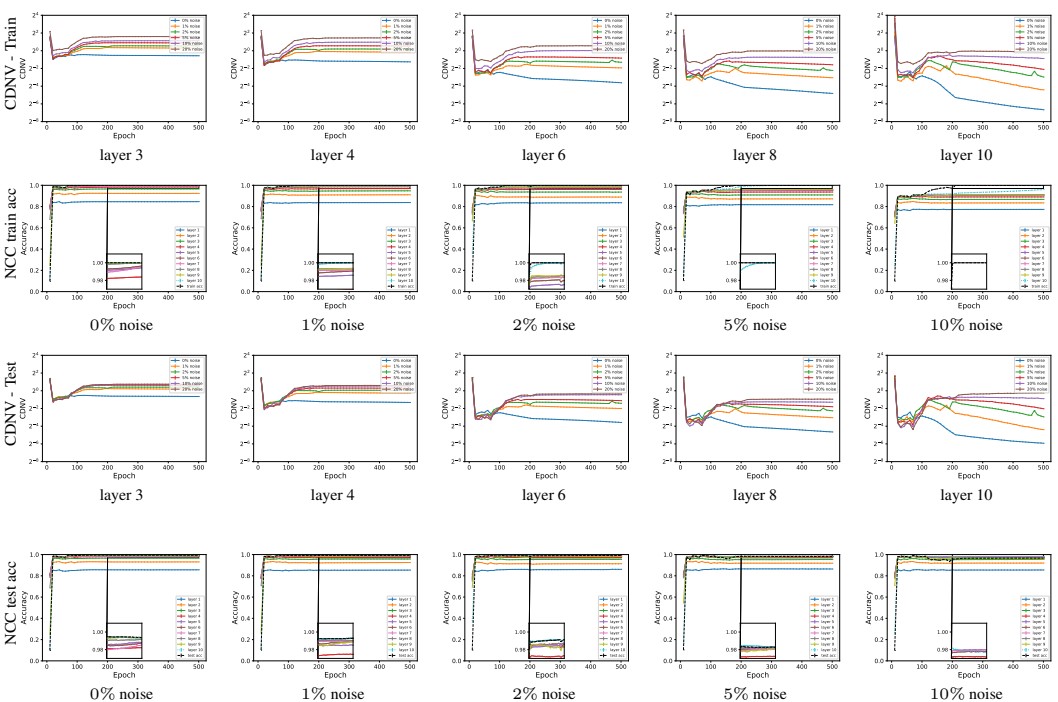

Figure 18: **Intermediate neural collapse of CONV-10-50 trained on MNIST with partially corrupted labels.** See Fig. 14 for details.

## B  PROOFS

**Proposition 1.** *Let $m \in \mathbb{N}$, $p \in (0, 1/2)$, $\alpha \in (0,1)$ and $\epsilon \in (0,1)$. Assume that the error of the learning algorithm is $\delta_m^1$-uniform. Assume that $S_1, S_2 \sim P_B(m)$. Let $h_{S_1}^\gamma$ be the output of the learning algorithm given access to a dataset $S_1$ and initialization $\gamma$. Then,*

$$
\begin{aligned}
\mathbb{E}_{S_1} \mathbb{E}_\gamma [\mathrm{err}_P(h_{S_1}^\gamma)] \;\leq\; & \mathbb{P}_{S_1, S_2, \tilde{Y}_2} \left[ \mathbb{E}_\gamma[\mathscr{d}_{S_1}^\epsilon(h_{S_1}^\gamma)] \;\geq\; \mathscr{d}_{\min}^\epsilon(\mathcal{G}, S_1 \cup \tilde{S}_2) \right] \\
& + (1+\alpha)p + \delta_m^1 + \delta_{m,p,\alpha}^2,
\end{aligned}
\tag{3}
$$

*where $\tilde{Y}_2 = \{\tilde{Y}_i\}_{i=1}^m$ is uniformly selected to be a set of labels that disagrees with $Y_2$ on $pm$ values.*

*Proof.* Let $S_1 = \{(x_i^1, y_i^1)\}_{i=1}^m$ and $S_2 = \{(x_i^2, y_i^2)\}_{i=1}^m$ be two balanced datasets. Let $\epsilon > 0$, $p > 0$ and $q = (1 + \alpha)\, p$. Let $\tilde{Y}_2$ and $\hat{Y}_2$ be a uniformly selected set of labels that disagree with $Y_2$ on $pm$ and $qm$ randomly selected labels (resp.). We denote by $\tilde{S}_2$ and $\hat{S}_2$ the relabeling of $S_2$ with the labels in $\tilde{Y}_2$ and in $\hat{Y}_2$ (resp.). We define four different events,

$$
\begin{aligned}
A_1 \;=\; & \{(S_1, S_2, \tilde{Y}_2) \mid \exists\, q \geq (1+\alpha)\, p : \; \mathscr{d}_{\min}^\epsilon(\mathcal{G}, S_1 \cup \tilde{S}_2) > \mathbb{E}_{\hat{Y}_2}[\mathscr{d}_{\min}^\epsilon(\mathcal{G}, S_1 \cup \hat{S}_2)]\} \\
A_2 \;=\; & \{(S_1, S_2) \mid \text{the mistakes of } h_{S_1}^\gamma \text{ are not uniform over } S_2\} \\
A_3 \;=\; & \{(S_1, S_2, \tilde{Y}_2) \mid (S_1, S_2, \tilde{Y}_2) \notin A_1 \cup A_2 \text{ and } \mathbb{E}_\gamma[\mathscr{d}_{S_1}^\epsilon(h_{S_1}^\gamma)] < \mathscr{d}_{\min}^\epsilon(\mathcal{G}, S_1 \cup \tilde{S}_2)\} \qquad (6) \\
A_4 \;=\; & \{(S_1, S_2, \tilde{Y}_2) \mid (S_1, S_2, \tilde{Y}_2) \notin A_1 \cup A_2 \text{ and } \mathbb{E}_\gamma[\mathscr{d}_{S_1}^\epsilon(h_{S_1}^\gamma)] \geq \mathscr{d}_{\min}^\epsilon(\mathcal{G}, S_1 \cup \tilde{S}_2)\} \\
B_1 \;=\; & \{(S_1, S_2, \tilde{Y}_2) \mid \mathbb{E}_\gamma[\mathscr{d}_{S_1}^\epsilon(h_{S_1}^\gamma)] \geq \mathscr{d}_{\min}^\epsilon(\mathcal{G}, S_1 \cup \tilde{S}_2)\}
\end{aligned}
$$

By the law of total expectation

$$
\begin{aligned}
\mathbb{E}_{S_1} \mathbb{E}_\gamma [\mathrm{err}_P(h_{S_1}^\gamma)] \;=\; & \mathbb{E}_{S_1, S_2} \mathbb{E}_\gamma [\mathrm{err}_{S_2}(h_{S_1}^\gamma)] \\
=\; & \sum_{i=1}^4 \mathbb{P}[A_i] \cdot \mathbb{E}_{S_1, S_2, \tilde{Y}_2} [\mathbb{E}_\gamma[\mathrm{err}_{S_2}(h_{S_1}^\gamma)] \mid A_i] \qquad (7) \\
\leq\; & \mathbb{P}[A_1] + \mathbb{P}[A_2] + \mathbb{E}_{S_1, S_2, \tilde{Y}_2}[\mathbb{E}_\gamma[\mathrm{err}_{S_2}(h_{S_1}^\gamma)] \mid A_3] + \mathbb{P}[B_1],
\end{aligned}
$$

where the last inequality follows from $\mathrm{err}_{S_2}(h_{S_1}^\gamma) \leq 1$, $\mathbb{P}[A_3] \leq 1$ and $A_4 \subset B_1$.

We would like to upper bound each one of the above terms. First, we notice that since the mistakes of the network are $\delta_m^1$-uniform, $\mathbb{P}[A_2] \leq \delta_m^1$. In addition, by definition $\mathbb{P}[A_1] \leq \delta_{m,p,\alpha}^2$.

As a next step, we upper bound $\mathbb{E}_{S_1, S_2, \tilde{Y}_2}[\mathbb{E}_\gamma[\mathrm{err}_{S_2}(h_{S_1}^\gamma)] \mid A_3]$. Assume that $(S_1, S_2, \tilde{Y}_2) \in A_3$. Hence, $(S_1, S_2, \tilde{Y}_2) \notin A_1 \cup A_2$. Then, the mistakes of $h_{S_1}^\gamma$ over $S_2$ are uniformly distributed (with respect to the selection of $\gamma$). Assume by contradiction that $\mathrm{err}_{S_2}(h_{S_1}^\gamma) > (1 + \alpha)\, p$ with non-zero probability over the selection of $\gamma$. Then, since the mistakes of $h_{S_1}^\gamma$ over $S_2$ are uniformly distributed, $\mathrm{err}_{S_2}(h_{S_1}^\gamma) > (1 + \alpha)\, p$ for all initializations $\gamma$. Therefore, we have

$$
\mathbb{E}_{\hat{Y}_2}[\mathscr{d}_{\min}^\epsilon(\mathcal{G}, S_1 \cup \hat{S}_2)] \;\leq\; \mathbb{E}_\gamma[\mathscr{d}_{S_1}^\epsilon(h_{S_1}^\gamma)] \;<\; \mathscr{d}_{\min}^\epsilon(\mathcal{G}, S_1 \cup \tilde{S}_2),
$$

where the first inequality follows from the definition of $\mathscr{d}_{\min}^\epsilon(\mathcal{G}, S_1 \cup \hat{S}_2)$ and the second one by the assumption that $(S_1, S_2, \tilde{Y}_2) \in A_3$. However, this inequality contradicts the fact that $(S_1, S_2, \tilde{Y}_2) \notin A_1$. Therefore, we conclude that in this case, $\mathbb{E}_\gamma[\mathrm{err}_{S_2}(h_{S_1}^\gamma)] \leq (1 + \alpha)\, p$ and $\mathbb{E}_{S_1, S_2, \tilde{Y}_2}[\mathbb{E}_\gamma[\mathrm{err}_{S_2}(h_{S_1}^\gamma)] \mid A_3] \leq (1 + \alpha)\, p$. $\qquad\square$

**Proposition 2.** *Let $m \in \mathbb{N}$, $p \in (0, 1/2)$, $\alpha \in (0,1)$ and $\epsilon \in (0,1)$. Assume that the error of the learning algorithm is $\delta_m^1$-uniform. Let $S_1, S_2, S_1^i, S_2^i \sim P_B(m)$ (for $i \in [k]$). Let $\tilde{Y}_2^i = \{\tilde{y}_i\}_{i=1}^m$ be a set of labels that disagrees with $Y_2^i$ on uniformly selected $pm$ labels and $\tilde{S}_2^i$ is a relabeling of $S_2$ with the labels in $\tilde{Y}_2^i$. Let $h_{S_1}^\gamma$ be the output of the learning algorithm given access to a dataset $S_1$ and initialization $\gamma$. Then, with probability at least $1 - \delta$ over the selection of $\{(S_1^i, S_2^i, \tilde{Y}_2^i)\}_{i=1}^k$, we*

*have*

$$\mathbb{E}_{S_1}\mathbb{E}_\gamma[\mathrm{err}_P(h_{S_1}^\gamma)] \;\leq\; \frac{1}{k}\sum_{i=1}^{k}\mathbb{I}\left[\mathbb{E}_\gamma[\ell^\epsilon_{S_1^i}(h_{S_1^i}^\gamma)] \;\geq\; \ell^\epsilon_{\min}(\mathcal{G}, S_1^i \cup \tilde{S}_2^i)\right]$$

$$+ (1+\alpha)\,p + \delta_m^1 + \delta_{m,p,\alpha}^2 + \sqrt{\frac{\log(2/\delta)}{2k}}.$$

*Proof.* By Prop. 1, we have

$$\mathbb{E}_{S_1}\mathbb{E}_\gamma[\mathrm{err}_P(h_{S_1}^\gamma)] \;\leq\; \mathbb{P}_{S_1,S_2,\tilde{Y}_2}\left[\mathbb{E}_\gamma[\ell^\epsilon_{S_1}(h_{S_1}^\gamma)] \;\geq\; \ell^\epsilon_{\min}(\mathcal{G}, S_1 \cup \tilde{S}_2)\right]$$

$$+ (1+\alpha)\,p_m + \delta_m^1 + \delta_{m,p,\alpha}^2$$

We define i.i.d. random variables

$$V_i \;=\; \mathbb{I}\left[\mathbb{E}_\gamma[\ell^\epsilon_{S_1^i}(h_{S_1^i}^\gamma)] \;\geq\; \ell^\epsilon_{\min}(\mathcal{G}, S_1^i \cup \tilde{S}_2^i)\right]. \tag{8}$$

Therefore, we can rewrite,

$$\mathbb{P}_{S_1,S_2,\tilde{Y}_2}\left[\mathbb{E}_\gamma[\ell^\epsilon_{S_1}(h_{S_1}^\gamma)] \;\geq\; \ell^\epsilon_{\min}(\mathcal{G}, S_1 \cup \tilde{S}_2)\right] \;=\; \mathbb{E}[V_1] \tag{9}$$

By Hoeffding's inequality,

$$\mathbb{P}\left[\left|k^{-1}\sum_{i=1}^{k}V_i - \mathbb{E}[V_1]\right| \;\geq\; \epsilon\right] \;\leq\; 2\exp(-2k\epsilon^2). \tag{10}$$

By choosing $\epsilon = \sqrt{\log(1/2\delta)/2k}$, we obtain that with probability at least $1-\delta$, we have

$$\mathbb{E}[V_1] \;\leq\; \frac{1}{k}\sum_{i=1}^{k}V_i + \sqrt{\log(1/2\delta)/2k}. \tag{11}$$

When combined with Prop. 1, we obtain the desired bound. $\qquad\square$

