# OpenReview forum: "On the Implicit Bias Towards Depth Minimization in Deep Neural Networks"
_ICLR.cc/2023/Conference — Submitted to ICLR 2023_

### Official Review · Reviewer_VKrE · 2022-10-24

**Confidence:** 2
**Clarity, Quality, Novelty And Reproducibility:** The paper is well-written and easy to…
**Correctness:** 3
**Technical Novelty And Significance:** 3
**Empirical Novelty And Significance:** 3
**Recommendation:** 5

**Strength And Weaknesses:**

Strength:
- Several insightful phenomena are observed (or validated): the network collapses until very late. The collapse in the middle layers matters for generalizations.
- A theoretical generalization bound in effective depth is proposed, the bound can serve as a good indicator for MLPs and ConvNets.

Weakness:
- The experiments are only done for ConvNets and MLPs, how does this estimation work for more practically used architectures such as ResNets or Transformers?
- The plots are very difficult to read because they are way too small.
- The author did not provide enough support on why the current bound is better than what have defined before.

**Summary Of The Paper:**

The paper proposes to understand the generalization of deep neural networks. It characterizes the effective depth as the first layer for which sample embeddings are separable using the nearest-class center classifier. Then a generalization bound based on the effective depth is suggested as an indicator of generalization performance. Some experiments are conducted to validate the effectiveness of this bound in estimating the generalization.




**Summary Of The Review:**

This work points to a new angle of understanding the generalization performance -- how neural collapse in the middle layer affects generalization. The neural collapse in the middle layer is further quantified by the "effective depth". The only concern that comes from the reviewer is the gap between theory and practice. It would be nice to see some experiential results on more "advanced" neural networks.

---

> ### Author Response · Authors · 2022-11-17
> **Author response**
>
> The experiments are only done for ConvNets and MLPs, how does this estimation work for more practically used architectures such as ResNets or Transformers?
>
> > Following the reviews, we added new experiments with the residual convolutional architecture CONVRES-L-H (see their description in Sec. 4.1 and Appendix A.1).
>
> > 1. In Tab. 2 we added the result of estimating the bound for a CONVRES-10-50  trained on CIFAR10. We obtained a bound of 0.4 on a 0.29 test error which is a remarkably tight result.
> > 2. In Figs. 4 and Fig. 5 we reproduce the observations in paragraph “Intermediate neural collapse” in Sec. 4.2 for the CONVRES-L-H architectures, which is a residual convolutional network with $L$ residual blocks. This architecture is described in detail in Appendix A.1.
> > 3. In Fig. 15 we reproduce the results in paragraph “NCC separability with partially corrupted labels” in Sec. 4.2 for the CONVRES-10-500 architecture. As can be seen, when increasing the number of noisy labels, the degrees of NCC separability across the layers tend to decrease.
>
> > Regarding Transformers, we are actively generating experiments with ViTs and have promising preliminary results. However, due to time and resource constraints, we were unable to provide concrete bounds by the deadline.
>
> The plots are very difficult to read because they are way too small.
>
> > Following the reviews, we increased the size of the plots in the main text. For many of the experiments in the appendix, we added summarized bigger plots (see Figs. 6,8,10,12).
>
> The author did not provide enough support on why the current bound is better than what have defined before.
>
> > To motivate our new approach, we added in-depth discussions on the inherent flaws of classic generalization bounds in Sections 1 and 3.2. We also discuss the benefits of our generalization bound in Sec. 3.2.
>
> To support our claim that our bound is better than previous bounds, we conducted several experiments comparing the bounds (see paragraph “Comparing our bound with standard generalization bounds” in Sec. 4.2). In Tab. 1 (previously Tab. 2) we report the averaged test error of the networks (the fourth row), our bound (sixth row), and the baseline bounds (seventh row and beyond). The comparison is done with various architectures of varying depths and multiple datasets. As can be seen, our bound consistently provides non-vacuous estimates of the test error (less than 1), whereas all baseline bounds across all experiments are greater than 3500. Therefore, the baseline bounds are meaningless in practice. To make the presentation of Tab. 1 clearer we updated its caption and moved it to the second page.
>
> As shown in Tab. 1, the baseline bounds rapidly increase with depth. For example, the Spectral-Frobenius bound of Neyshabur et al. 2019 equals 9.4833e+04 for a 16 layers network and equals 1.033e+08 for a 20 layers network. In contrast, both the test error and our bound remain fairly constant when varying the depth. To emphasize this point, we added a new paragraph in Section 3.2 discussing why our bound is not expected to grow when increasing $L$.

---

### Official Review · Reviewer_JHSQ · 2022-10-24

**Confidence:** 3
**Correctness:** 2
**Technical Novelty And Significance:** 3
**Empirical Novelty And Significance:** 3
**Recommendation:** 3

**Clarity, Quality, Novelty And Reproducibility:**

Clarity and quality is not the strength of this paper as plots are ridiculously small and one can barely guess what they show. Reproducibility I did not see a link to the code used to produce the values in tables and figures.

**Strength And Weaknesses:**

Strengths:
- I kind of like the idea of evaluating neural collapse for measuring generalisation.
- Mathematical analysis seems interesting (although I did not check it in detail).

Weakneses:
- I am not convinced that this is better measure than what is already present as there is real no proof of that. In the introduction and other parts it states "In many practical settings [...] making the bounds vacuous". This is repeated several times, but I failed to see an in-depth analysis of those "practical settings" and what are those "vacuous bounds". One finds something at the end of section 3 but is the same repetition about vacuous bounds and sqrt(m), and not an in-depth analysis in how it compares to the other "classical measures". It all becomes very vague and repetitive when addressing the advantages of the presented measure when compared to those "classical" ones.
- All plots are awful as they are barely visible. Labels must be font size 2.

**Summary Of The Paper:**

This paper proposes a method for evaluation the generalising capabilities of neural networks through neural collapse

**Summary Of The Review:**

I have mixed feelings. I am up for new ideas, but the paper does not do a good job on:

 1. Showing that is better measure than other measures. Instead of repeating over and over that "classical measures" provide "vacuous bounds", how about using that space for and in-depth analysis and comparison with each one of those measures?. and

2. I do not think authors spent the time necessary on the figures so that the reader would not need a microscope to see what is plotted there.

---

> ### Author Response · Authors · 2022-11-17
> **Author response**
>
> I am not convinced that this is better measure than what is already present as there is real no proof of that. In the introduction and other parts it states "In many practical settings [...] making the bounds vacuous". This is repeated several times, but I failed to see an in-depth analysis of those "practical settings" and what are those "vacuous bounds". One finds something at the end of section 3 but is the same repetition about vacuous bounds and sqrt(m), and not an in-depth analysis in how it compares to the other "classical measures". It all becomes very vague and repetitive when addressing the advantages of the presented measure when compared to those "classical" ones.
>
> > Following this comment, we added in-depth discussions on the inherent flaws of classic generalization bounds in Sections 1 and 3.2. In Sec. 3.2 we also discuss the advantages of our generalization bound.
> We would like to mention that directly comparing different bounds theoretically is difficult unless two bounds are based on very similar derivations. (1) Generalization bounds rely on different notions of complexity that are usually incomparable (e.g., number of parameters, norms, our effective depth, etc). (2) These quantities are usually dependent on many factors (e.g., the data, architecture, and optimizer) and it is often difficult to predict which bound would be better in practice. As a result, the most effective way to determine which bound is useful is by comparing them empirically.
>
> > To support our claims, we conducted several experiments comparing the bounds (see paragraph “Comparing our bound with standard generalization bounds” in Sec. 4.2). In Tab. 1 (previously Tab. 2) we report the averaged test error of the networks (the fourth row), our bound (sixth row), and the baseline bounds (seventh row and beyond). The comparison is done with various architectures of varying depths and multiple datasets. As can be seen, our bound consistently provides non-vacuous estimates of the test error (less than 1), whereas all baseline bounds across all experiments are greater than 3500 (!). Therefore, the baseline bounds are meaningless in practice. To make the presentation of Tab. 1 clearer we updated its caption and moved it to the second page.
>
> > As shown in Tab. 1, as a general tendency the baseline bounds rapidly increase with depth. For example, the Spectral-Frobenius bound of Neyshabur et al. 2019 equals 9.4833e+04 for a 16 layers network and equals 1.033e+08 for a 20 layers network. In contrast, both the test error and our bound remain fairly constant when varying the depth. To emphasize this point, we added a new paragraph in Section 3.2 discussing why our bound is not expected to grow when increasing $L$.
>
> All plots are awful as they are barely visible. Labels must be font size 2.
>
> > Following the reviews, we increased the size of the plots in the main text. For many of the experiments in the appendix we added summarized bigger plots (see Figs. 6,8,10,12).
>
> Clarity and quality is not the strength of this paper as plots are ridiculously small and one can barely guess what they show. Reproducibility I did not see a link to the code used to produce the values in tables and figures.
>
> > Following the comment, we added our code to the supplementary material with the relevant settings to reproduce the experiments in the paper.

---

### Official Review · Reviewer_nZgi · 2022-11-02

**Confidence:** 4
**Clarity, Quality, Novelty And Reproducibility:** The paper is clear and novel, the wri…
**Correctness:** 4
**Technical Novelty And Significance:** 3
**Empirical Novelty And Significance:** 2
**Recommendation:** 6

**Strength And Weaknesses:**

Strength:
- The idea of establishing a generalization bound based on the neural collapsing phenomenon is novel and interesting.
- The experiments are comprehensive and details are provided.

Weakness:
- It's unclear whether some of the technical settings are necessary. For example, I don't see how the specific structures of Conv-L-H and MLP-L-H come into play in the theory.
- I somehow feel the first term in the bound could be pretty large, hence making the bound vacuous. The authors provide some experiments to justify that the term is small, however, the way they compute d_min is by relying on the implicit bias of SGD, which is really not reliable since SGD may not be able to find really the best shallow network that fits the data. I wonder we can find a smaller d_min with the following algorithm: given some target depth d, we train a (d-1)-layer NN classifier to achieve less than eps error, then we embed the last linear layer into an additional layer by add a bunch of 0 weights. Would this lead to a d-layer NN with epsilon error? If so I feel this might give us a smaller depth than train d+1 layer NN and then chop off the last linear layer.

**Summary Of The Paper:**

This paper proposes a new generalization bound for deep neural networks based on the effective depth. The idea is motivated by the empirical observation that neural networks’ representations exhibit neural collapsing. The authors define the notions of effective depth and minimal depth of a neural network, and establish a generalization bound based on the comparison between them. The paper also provides empirical estimation of the bound which is shown to be tighter than previous bounds.

**Summary Of The Review:**

I think this paper provides a novel perspective of generalization bounds from the angle of neural collapsing. Although the theory is not complicated, it's idea is novel as far as I can tell, and could be of interest to many theoreticians in the ML community. Thus I would be happy to see this work getting accepted.

---

> ### Author Response · Authors · 2022-11-17
> **Author response**
>
> It's unclear whether some of the technical settings are necessary. For example, I don't see how the specific structures of Conv-L-H and MLP-L-H come into play in the theory.
>
> > Following the reviews, we shortened the architecture descriptions and moved them to the experimental setup. Appendix A.1 now includes a more detailed description of the architectures. We also added a new architecture denoted by CONVRES-L-H since we added experiments with residual convolutional networks.
>
> > Regarding the remaining details (for example, losses, optimizers, regularization, and so on), we used Sec. 2 to cover the details for both theory and experiments. Even though some details are unrelated to the proof of Prop. 1, since Prop. 1 is a bit abstract, they provide context for when we expect it to be useful.
>
> I somehow feel the first term in the bound could be pretty large, hence making the bound vacuous. The authors provide some experiments to justify that the term is small, however, the way they compute d_min is by relying on the implicit bias of SGD, which is really not reliable since SGD may not be able to find really the best shallow network that fits the data. I wonder we can find a smaller d_min with the following algorithm: given some target depth d, we train a (d-1)-layer NN classifier to achieve less than eps error, then we embed the last linear layer into an additional layer by add a bunch of 0 weights. Would this lead to a d-layer NN with epsilon error? If so I feel this might give us a smaller depth than train d+1 layer NN and then chop off the last linear layer.
>
> > We repeated the experiment performed on MNIST in Tab. 2 (training a CONV-10-50 on MNIST) while estimating d_min with the algorithm suggested by the reviewer. We got the same result as we previously had (a bound of 0.1). We were unable to provide additional results by the deadline due to time and resource constraints; however, we are working on reproducing the results for additional cases.

---

### Author Response · Authors · 2022-11-18
**Summary of changes**

We would like to thank the reviewers for taking the time to read our paper, and providing useful feedback for our work. Following the reviewers, we made the following changes:
1. Following the comment of VKrE, we added a new estimation of our bound for residual convolutional networks (see Tab. 2).
2. Following the comment of VKrE, we added new results validating the minimal depth hypothesis for residual convolutional networks (see Figs. 4 and 5).
3. Following the comment of VKrE, we added new results comparing the intermediate layers NCC separability rates of networks trained with varying degrees of label noise (see Fig. 11).
4. Following the comments of JHSQ and VKrE, we added a detailed review of the inherent flaws in traditional generalization bounds in Sections 1 and 3.2.
5. Following the comments of JHSQ and VKrE, we extended the discussion of the advantages of our bound in Section 3.2, including an explanation of why it is not expected to increase when varying the network’s depth $L$.
6. To highlight the empirical comparison between our bound and well-established bounds in the literature, we moved Tab. 1 (previously Tab. 2) to the second page. We also rewrote its caption.

---

### Author Response · Authors · 2022-12-01
**We would be happy to address any follow-up questions**

We thank the reviewers for the detailed feedback and useful ideas. As detailed in the posted summary of changes, we made an effort to factually address the main concerns.

We would appreciate the opportunity to discuss our work further if the response to each reviewer has not already addressed all concerns.

---

### Author Response · Authors · 2022-12-08
**Final Request for Reviewer Feedback**

We thank the reviewers for the detailed feedback for our paper.

We have invested a very large amount of time addressing every claim that was made in the responses. We additionally revised the manuscript, and added the code for reproducibility. We believe the results justify a second judgment, or at the very least some input as to why you believe these results do not address your claims. This will be incredibly helpful for a constructive reviewing process, and given that the discussion period is nearly over, we would greatly appreciate input from any one of the reviewers. We thank you for your time, and appreciate any feedback you can give us!

Thank you,
The Authors

---

### Decision · Program_Chairs · 2023-01-20

**Decision:**

Reject

**Justification For Why Not Higher Score:**

The presentation of the main results is too implicit, and the figures and tables are borderline unreadable without constant zooming.  The submission simultaneously doesn't say enough explicitly in the discussion of results, and is unclear in the figures presenting the results.  The results are indeed important and good, but need a substantially improved presentation to have an appropriate impact.

**Justification For Why Not Lower Score:**

N/A

**Metareview: Summary, Strengths And Weaknesses:**

The submission proposes a very interesting theoretical analysis of the implicit bias of SGD to produce networks with low effective depth.  They subsequently develop a generalization bound that appears to have strongly improved performance over those based on uniform convergence bounds, giving non-trivial generalization bounds when others are in practice completely vacuous.  These results are interesting, and I strongly believe they should in the end be published in a good quality ML venue.  I have concerns, though, about the current presentation.  The reviewers were unanimous in their opinion that the clarity of the submission is lacking, and that improvement over classical results was not sufficiently demonstrated.  With the rebuttal text, and additional results, this becomes more convincing, but the presentation in the paper is not very clear, and the significance not clearly stated.  The real results are hidden in tiny figures with tiny font, and not stated sufficiently clearly in the results and discussion, although the background results in the introduction are improved.  The paper has improved over the initial submission, but needs a thorough revision, and clear statement of improvement in bounds that has appropriate emphasis and discussion in the results, discussion and conclusions of the paper.  Making clear figures to illustrate this may not fit in the page limits for a conference like ICLR, and so either the focus of the paper might need to change to fit in a conference format, or a journal might need to be considered.

Theoretical justification of architecture choices missing, this can be OK, but the question from the first reviewer wasn't really answered in the rebuttal.